evolution/genomics

adaptive walks, punctuated equilibrium, rate of evolution, positive selection

**Author for correspondence:**
Anastasia V. Stolyarova
e-mail: anastasia.v.stolyarova@gmail.com

# Bursts of amino acid replacements in protein evolution

Anastasia V. Stolyarova[1], Georgii A. Bazykin[1,2], Tatyana V. Neretina[3,4] and Alexey S. Kondrashov[4,5]

[1]Skolkovo Institute of Science and Technology, Skolkovo 143026, Russia
[2]Institute for Information Transmission Problems (Kharkevich Institute) of the Russian Academy of Sciences, Moscow 127994, Russia
[3]White Sea Biological Station, Biological Faculty, M. V. Lomonosov Moscow State University, Leninskie Gory, Moscow 119991, Russia
[4]Department of Bioengineering and Bioinformatics, M. V. Lomonosov Moscow State University, Moscow 119234, Russia
[5]Department of Ecology and Evolutionary Biology, University of Michigan, 830 North University, Ann Arbor, MI 48109-1048, USA

AVS, 0000-0002-1418-6673

Evolution can occur both gradually and through alternating episodes of stasis and rapid changes. However, the prevalence and magnitude of fluctuations of the rate of evolution remain obscure. Detecting a rapid burst of changes requires a detailed record of past evolution, so that events that occurred within a short time interval can be identified. Here, we use the phylogenies of the Baikal Lake amphipods and of Catarrhini, which contain very short internal edges which make this task feasible. We detect six salient bursts of evolution of individual proteins during such short time periods, each involving between six and 38 amino acid substitutions. These bursts were extremely unlikely to have occurred neutrally, and were apparently caused by positive selection. On average, in the course of a time interval required for one synonymous substitution per site, a protein undergoes a strong burst of rapid evolution with probability at least approximately 0.01.

## 1. Introduction

(Non)uniformity of the rate of evolution is one of the oldest and most contentious issues in evolutionary biology. On the one hand, at both molecular and morphological levels, evolution often occurs gradually, so that its rate is approximately constant [1–5]. In particular, this is the case for selectively neutral segments of genomes, which evolve at rates equal to the

corresponding mutation rates ('molecular clock' [6,7]). Instances of gradual adaptive evolution are also known [8,9].

On the other hand, evolution may also occur mostly through short bursts of changes alternating with long periods of stasis ('punctuated gradualism' or 'punctuated equilibrium' [10,11]). Examples of punctuated equilibrium are provided by the evolution of mammalian body weight [12], hominoid body size [13], several morphological traits of rockfish [14], intersexual signalling of cranes [15] and many other data [16–22]. Clearly, both gradual and burst-like evolution does happen, but their relative importance remains controversial [23–26]. Of course, there can be many causes for alternating episodes of stasis and evolution and of punctuation in molecular and morphological evolution.

In order to detect short bursts of changes, one needs to be able to identify evolutionary events that occurred during a short interval of time. This is easy to accomplish if a very detailed palaeontological record is available, such as those that exist for some marine invertebrates, e.g. Foraminifera [27]. However, such records are exceptions rather than the rule. Furthermore, palaeontological data usually shed light only on the morphology of organisms and, thus, cannot reveal bursts of changes at the level of genomes. Fortunately, it may be possible to identify such bursts indirectly, through comparison of genomes of extant species, as long as their phylogenetic tree contains very short internal edges. Unfortunately, despite an avalanche of genomic data, the vast majority of the currently available phylogenetic trees do not satisfy this requirement. Still, we took advantage of two phylogenetic trees that contain such edges, those of the Lake Baikal amphipods [28] and of Catarrhini [29], UCSC 100 vertebrates multiple alignment, and investigated short bursts in the evolution of their proteins.

# 2. Material and methods

## 2.1. Dataset

We used two datasets. First, we considered transcriptomes-based clusters of orthologous genes (COGs) with exactly one orthologue represented in each species for five clades of the Lake Baikal amphipods (gammarids) (64 species and 3399 COGs in total), and the phylogeny based on them. The size of a clade varies from 6 to 24 species (electronic supplementary material, figure S1A). Each clade consists of closely related species, which makes it possible to reconstruct ancestral states with high confidence. Thus, all amino acid substitutions that constitute a burst can be reliably identified. The search for bursts was performed for each clade separately.

Second, we considered a multiple alignment of protein-coding genes from 11 primates species obtained from the 100 vertebrates genomes alignment of the UCSC Genome Browser together with the corresponding reconstructed phylogenetic tree (electronic supplementary material, figure S1B). In total, there were 17 755 alignments of protein-coding genes of primates containing columns without gaps.

Only internal edges, i.e. segments of the phylogenetic tree ancestral to more than one species, were used in our analysis. For both datasets, we only considered internal edges of length less than 0.005 dS units. We used *codeml* program of the PAML package [30] to reconstruct substitution histories of sequences and to estimate dN/dS values. Only gapless alignment columns were considered.

Presumptive functions of amphipod genes were inferred from blast2GO predictions [28] and from the genome annotation of a related species *Hyalella azteca* [31]; functions of primate genes were inferred the human genome annotation (hg38).

## 2.2. Analysis

We assumed the neutral null model (dN = dS), which makes it possible to detect only the longest and the fastest bursts (see Discussion). We related dN of the gene on a particular edge of the phylogenetic tree to the length of this edge. Length edge was measured in the units of dS on the basis of all the available genes. We used this approach, instead of considering the dS value of only the gene that underwent a burst, because it is impossible to estimate the dS for an individual gene on a short edge with precision. For example, the expected number of synonymous substitutions in a gene encoding a 200 amino acid long protein on the edge of length 0.005 dS is only approximately 1. For a particular edge, the *p*-value associated with a non-synonymous burst was calculated for each gene as the probability of observing this many or more non-synonymous substitutions in a Poisson distribution with the

parameter equal to the edge length (dS) (or, for bursts spanning multiple edges in a row, the sum of their lengths) multiplied by the number of non-synonymous sites in the gene (estimated according to [32]). The obtained $p$-values were adjusted for multiple comparisons using the Benjamini–Hochberg method.

As a control, we also searched for statistically significant bursts of synonymous substitutions on the same set of short edges using an analogous approach.

## 2.3. Filtering

The primary set of putative bursts with Benjamini–Hochberg adjusted $p$-values $< 0.05$ was filtered as follows.

First, alignments containing more than 50% of columns with gaps were excluded. Second, to ensure the precise phylogenetic positioning of all substitutions that constituted a burst, we excluded sites with *codeml* posterior probabilities for the reconstructed ancestral variant less than 0.8 (which will be the case for discordant genes, e.g. caused by incomplete lineage sorting) and recalculated the statistics for this gene.

Third, to safeguard against contribution of anciently divergent paralogues or pseudogenes rather than orthologues to our findings, we applied additional filtering. We required that the dS value that characterized the edge of the putative burst obtained using the considered gene was lower than the dS value for this edge obtained using all genes (adjusted $p$-value $> 0.001$). Genes with substitutions in multiple repeated domains were excluded. Next, we required the absence of evidence for paralogues or duplications of the considered gene as follows. For each gene, we determined the pre-burst sequence as the reconstructed sequence of the phylogenetic node immediately ancestral to the burst-carrying edge(s), and the post-burst sequence as the reconstructed sequence of the phylogenetic node immediately descendant to it. For gammarids, we mapped raw transcriptomic reads of the considered gene from all species onto the pre-burst and post-burst sequences. If any reads from any of the species descendant to the edge of the provisional burst supported the pre-burst variant, or if any reads from any of the species not descendant to the edge of the provisional burst supported the post-burst variant, this gene was discarded. For primates, we aligned the pre-burst and post-burst sequences onto the assembled genomes of this gene from all species, and proceeded analogously. For primates, we also required that the genomic position of each burst-carrying gene was conserved in all primate genomes.

Finally, the burst-containing alignments that survived these filters were curated manually for any evidence for alignment errors, low complexity and unexpected patterns in substitutions. If most substitutions constituting a burst fell into regions of poor alignment or were located in the very beginning or the very end of the gene, the corresponding putative burst was discarded.

## 3. Results

We searched for bursts of amino acid substitutions (bursts) within internal edges of phylogenetic trees that are shorter than 0.005 dS. Suitable edges are present in five clades of the phylogenetic tree of gammarids from Lake Baikal [28]: *Eulimnogammarus* and related genera (18 edges), *Pallasea* and related genera (10), *Hyallelopsis* (3), Acanthogammaridae s. str. (7) and Micruropidae (4); as well as within the Catarrhini clade (3 edges) of the tree of vertebrates (electronic supplementary material, figure S1) [29]. A burst consists of several amino acid substitutions which occurred in a protein within such an edge or, perhaps, within several successive edges of combined length below 0.005 dS.

In gammarids, five statistically significant bursts occurred in five proteins within two clades. Three of them occurred over the time period corresponding to a short individual edge of the phylogeny, while the remaining two spanned two very short adjacent edges. In Catarrhini, there was one significant burst (table 1). Each burst consisted of between 6 and 38 amino acid substitutions, or between 6 and 39 non-synonymous substitutions (as some amino acid sites underwent multiple non-synonymous substitutions), scattered throughout the protein (electronic supplementary material, figure S2). All edges that harbour bursts have 100% bootstrap support. Unfortunately, there is no accepted dating for amphipods diversification, so we cannot estimate the time of the corresponding bursts with precision; all that we know is that they have occurred after Baikal has originated in the Miocene, i.e. less than 30 Ma. Genes that harboured bursts are enriched in proteins located in mitochondria: they constitute three of the five such genes, although only 14% of the initial set of COGs are annotated as

**Table 1.** Bursts of evolution in proteins of the Baikal gammarids and Catarrhini.

| clade | edge number | edge length (dS) | gene name (predicted) | description of the protein | overall dN/dS for the gene (excluding the edge with the burst) | substitutions during burst | | adjusted p-value |
|---|---|---|---|---|---|---|---|---|
| | | | | | | non-syn. | syn. | |
| *Pallasea* and related genera | 1 | 0.0040 | DNAJC11 | DnaJ-like protein subfamily c member 11 | 0.41 | 39 | 1 | $1.01 \times 10^{-25}$ |
| | 1 | 0.0040 | MRPL22 | mitochondrial ribosomal protein L22 | 0.55 | 10 | 1 | 0.012 |
| | 2 + 3 | 0.0007 + 0.0008 | NOP16 | nucleolar protein 16-like | 0.35 | 6 | 2 | 0.046 |
| *Eulimnogammarus* and related genera | 4 | 0.0011 | MRPS25 | mitochondrial ribosomal protein S25 | 0.31 | 6.5 | 0.5 | 0.0011 |
| | 5 + 6 | 0.0024 + 0.0010 | AKR1 | aldo-keto reductase | 0.57 | 10 | 1 | 0.032 |
| primates | 7 | 0.0043 | PKR | interferon-induced protein kinase R | 1.27 | 18 | 2 | 0.0010 |

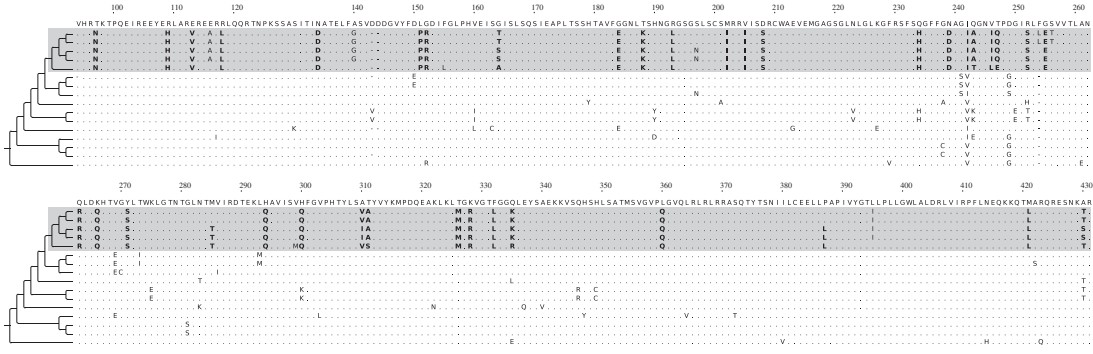

**Figure 1.** A fragment of the alignment of orthologous *DNAJC11* genes of *Pallasea* gammarids, which contains 39 non-synonymous substitutions on the internal edge 0.004 dS in length. Alleles derived in the adaptive burst are shown in bold. Total length of the alignment is 1677 nucleotides, or 1515 nucleotides without gaps.

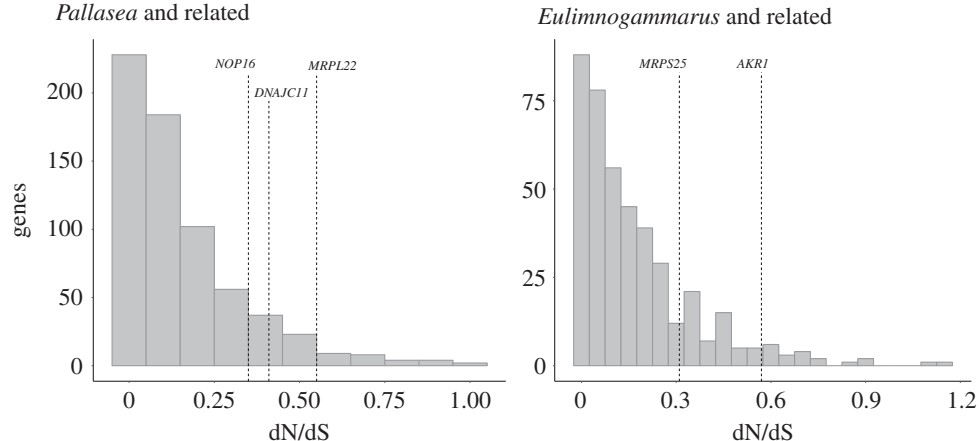

**Figure 2.** Distribution of dN/dS in genes of Baikal Amphipods in the clades carrying bursts. Genes with confirmed bursts are shown with dashed lines.

components of mitochondria (binomial test, *p*-value = 0.02; electronic supplementary material, figure S3) [28]. No significant synonymous bursts were observed.

The most remarkable burst involving 39 non-synonymous substitutions occurred in the mitochondrial chaperone gene (*DNAJC11*) on an edge of length 0.004 dS in the *Pallasea* clade (figure 1). This edge also harboured another burst in a protein located in mitochondria (L22 ribosome protein) (table 1). Bursts that are confined to one edge of the phylogenetic tree do not extend to preceding and/or successive edges (*p*-values for such edges greater than 0.27) (electronic supplementary material, figure S4). Hence, the characteristic duration of a burst is short, approximately $10^{-3}$ dS. The overall rate of evolution of some burst-carrying genes was somewhat higher than the average (table 1 and figure 2). Still, after multiple testing correction, there remains no genes with more than one statistically significant burst (*p*-values > 0.018, adjusted *p*-values = 1).

Phylogenetic tree of 11 species of Catarrhini has only three internal edges shorter than 0.005 dS (electronic supplementary material, figure S1B), and only one statistically significant burst has been detected. This small number may be due to several reasons: Catarrhini species are more distant from each other, which results in longer phylogenetic edges and less confident ancestral state reconstruction; moreover, a larger initial dataset leads to a more substantial multiple testing correction.

The detected burst occurred on the internal edge ancestral to two macaques species. Based on the divergence time estimates, we assume that the burst occurred at approximately 8–3 Ma [33]. The gene with the burst encodes the PKR protein (also known as EIF2AK), which is the eukaryotic translation initiation factor 2 kinase activated during viral infection. As in gammarids, the substitutions are scattered along the sequence (figure 3). Primate PKR contains two dsRNA binding motifs (DRBMs 1 and 2) and C-terminal catalytic kinase domain. The kinase domain carries 14 amino acid substitutions on the selected edge, and DRBM1 the remaining 4. Most substitutions lie in αD, αG and αH helices

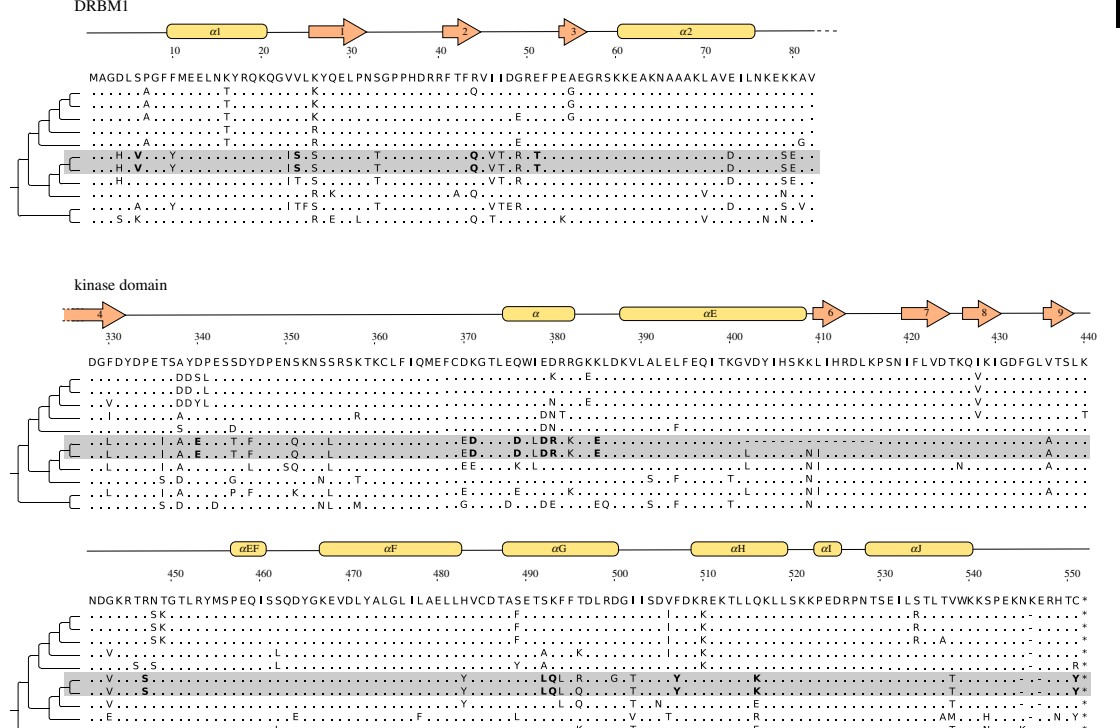

**Figure 3.** Alignment of *PKR* genes of Catarrhini (fragments) containing 18 non-synonymous substitutions on the internal edge ancestral to *Macaca mulatta* and *Macaca fascicularis*. The majority of substitutions (14) occurred in the kinase domain of the protein, the others fall in the dsRNA binding motif (DRBM1). Alleles derived in the adaptive burst are shown in bold.

or nearby, which have been shown to be enriched in positively selected sites [34]. αG helix and specifically positions with amino acid substitutions on the selected edge are involved in PKR interaction with eIF2α [35]; however, there is no evidence of bursts of evolution in *eIF2α* gene on the same edge.

# 4. Discussion

Allele replacements driven by positive selection are the fundamental genetic mechanism of adaptive evolution. These replacements can occur independently of each other or be correlated [36]. *A priori*, there is a continuum of possibilities, from fully independent individual substitutions to bursts of adaptive evolution, each consisting of multiple substitutions that occurred over a short period of time. We searched for such bursts within individual proteins, taking advantage of two phylogenetic trees, of the Lake Baikal gammarids [28] and of Catarrhini [29].

Using only internal edges is essential because in this case the derived sequence is observed in more than one species, so that rare sequencing and alignment errors would not lead to false discovery of bursts. Our criteria for detection of bursts were rather stringent: conservative filtering of alignments, a neutral null model (dN = dS) and multiple testing correction. Unfortunately, it is hard to define the correct null model which takes into account all possible features of protein evolution, for example, those resulting from non-uniformity of the mutation rate along the genomes. Thus, our *p*-values should be viewed with caution. Even neutrally evolving sequences can carry increased number of substitutions because of variation in the evolution rate over large time scales (overdispersed molecular clock [37–39]). However, the aim of our work was not to identify consistent deviations from the Poisson expectation, but to find the most radical outliers. The six bursts that we have found are likely to be 'real', in the sense of being caused by simultaneous or near-simultaneous action of positive selection at multiple sites within a protein.

Berglund *et al.* [40] and Galtier *et al.* [41] have used methods similar to ours to find episodes of accelerated evolution which could have non-adaptive explanations, such as gene conversion. By contrast, we used stringent criteria for detection of bursts, and those that we found are likely to occur

due to positive selection. In particular, the most radical bursts in Galtier *et al*. involve both synonymous and non-synonymous substitutions; while the bursts described in our paper, in particular in the novel amphipod dataset, are limited to non-synonymous sites. Unlike Galtier and Berglund, we see no GC bias in the bursts-composing substitutions: e.g. the number of AT → GC and GC → AT substitutions in the strongest burst are 14 and 21, correspondingly. This is inconsistent with gene conversion, and instead further supports the adaptive explanation.

Mutations that initiated amino acid substitutions that together constitute a burst are extremely unlikely to appear simultaneously as parts of one complex mutational event. Thus, a burst is likely to involve substitutions that were not precisely synchronous. In other words, a burst lasts longer than a substitution. Still, the bursts that we detected are quite short, at the evolutionary time scale. Indeed, four bursts were confined to just one internal edge shorter than 0.005 dS. The remaining two bursts, involving six and nine amino acid substitutions, each occurred on two successive internal edges, of lengths 0.0007 and 0.0008, and 0.0024 and 0.0010 dS, suggesting that approximately three to four non-synonymous substitutions occurred per 0.001 dS of evolutionary time. Among the substitutions involved in such composite bursts, neither occurred on both edges after a cladogenesis (multiple non-synonymous substitutions did not occur on edges leading to *Eulimnogammarus testaceus* or *Pallasea cancellus*).

Of course, the two phylogenetic trees which we studied almost certainly contained other bursts of positive selection-driven amino acid substitutions which we could not detect with certainty. This would be the case for any burst that occurred within an external edge of a tree, or within an internal edge that is not short enough, or even within a short internal edge as long as the burst itself involved only a small number of substitutions. Unfortunately, we cannot estimate the number of such real but not confidently detectable bursts.

Obviously, our ability to detect a burst depends on the length of the internal edge. Roughly speaking, all bursts that involve at least approximately eight amino acid substitutions within a protein of less than 300 amino acids can be detected within internal edges of length below 0.005 dS. Because we investigated 3411 proteins and the total length of all such edges in the gammarid tree was 0.15 dS, five bursts that we found in them imply that during an interval of time required for one synonymous substitution to occur per site, a protein undergoes a strong burst of adaptive evolution with probability approximately 0.01 (the estimate derived from primates is similar). If so, such bursts are not uncommon.

What can we say about the genes that underwent bursts? Not much: they evolve faster than an average gene, but only marginally so. Unexpectedly, three out of five genes encode proteins that are located in mitochondria: a mitochondrial chaperone and two mitochondrial ribosome proteins. This observation is hard to explain. Mitochondrial genomes of Baikal Lake gammarids have been shown to undergo intensive rearrangement, which in combination with mito-nuclear discordance and epistatic interactions between mitochondrial proteins coded in nuclear and mitochondrial genomes might lead to this phenomenon [42].

Because multiple nucleotide substitutions that constitute a burst occur very close to each other, making recombination between them negligible, Hill–Robertson interference [43] can be expected to impede their fixations. Let us consider the most extreme burst comprising 39 non-synonymous substitutions on the internal edge of length 0.004 dS. Assuming the per nucleotide per generation mutation rate is approximately $10^{-8}$, as in a number of animals [44], this edge corresponds to approximately 400 000 generations, leaving approximately 10 000 generations per each substitution, if they occurred without overlaps. Is this feasible? Every generation, $2N\mu$ mutations occur at a site, where $N$ is the census population size and $\mu$ is the mutation rate. An advantageous mutation will eventually reach fixation with probability $2sN_e/N$, where $N_e$ is the effective population size (see [45, p. 158]). Thus, the per generation probability of fixation of a particular advantageous mutation is $4N_e\mu s$. Under assumptions of $N_e = 10^5$ (limited data indicate nucleotide diversity approx. 0.01 in several Lake Baikal amphipods), $\mu = 10^{-8}$ and $s = 10^{-2}$, this probability becomes $4 \times 10^{-5}$, which is not very different from $10^{-4}$. Thus, successive accumulation of substitutions that constitute a burst, which makes them immune to the Hill–Robertson interference, cannot be ruled out. This would be especially the case if during the whole course of a burst selection favours mutations at all the 39 sites that constitute it, or, in other words, the order in which the substitutions occur is not prescribed [46]. If so, the target for advantageous mutation at a particular moment of time consists of all sites where substitutions did not yet occur, and their order depends on the order in which mutations appear.

Correlated positive selection at multiple sites that leads to a burst may emerge due to a variety of mechanisms. One possibility, of course, is a sudden, drastic change of the adaptive landscape of a protein. However, a burst can also occur as a result of only a small change of the landscape, if it caused a fold bifurcation which eliminated a fitness peak that was occupied by a protein [47, fig. 4].

This mechanism is compatible with the fact that all genes with bursts in gammarids show a low overall dN/dS ratio on the entire phylogenetic tree (less than 0.57), implying that these bursts of evolution affected genes that usually evolve slowly. By contrast, the *PKR* gene of primates possessed a high dN/dS ratio (greater than 1), implying that the burst in this gene involved an episode of additional acceleration of evolution which was generally fast. Hopefully, the number of available dense phylogenetic trees will soon become much larger, which will make it possible to study bursts of rapid evolution in more detail.

Data accessibility. Data available from the Dryad Digital Repository: https://doi.org/10.5061/dryad.40vp30d [48].
Authors' contributions. A.V.S. carried out the data analyses, participated in the design of the study and drafted the manuscript; G.A.B. participated in the design of the study and helped draft the manuscript; T.V.N. provided experimental validation of the results; A.S.K. conceived of the study and helped draft the manuscript. All authors gave final approval for publication.
Competing interests. The authors declare no competing interests.
Funding. This study was supported by Russian Science Foundation (grant no. 16-14-10173) to A.S.K.
Acknowledgements. We gratefully acknowledge Dr E. Braun and the anonymous reviewer whose comments helped to improve and to clarify the manuscript.

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
