## [Reviewer comments · Royal Society Open Science]

Review History

RSOS-181095.R0 (Original submission)

Review form: Reviewer 1 (William Amos)

Is the manuscript scientifically sound in its present form?

Yes

Are the interpretations and conclusions justified by the results?

Yes

Is the language acceptable?

Yes

Is it clear how to access all supporting data?

Not Applicable

Do you have any ethical concerns with this paper?

No

Have you any concerns about statistical analyses in this paper?

Yes

Recommendation?

Accept with minor revision (please list in comments)

Comments to the Author(s)

I rather like this paper though for me it tantalises more than it reveals. I feel it could be fleshed out a bit more, particularly with regard to the biggest 'hit' in point 3 below:

1. There is an assertion that these are the only sets of genomes that are suitable. This may be so, but it would be nice to hear the reasons why obvious candidates like *Drosophila* are not included.
2. I would like to see more justification for the size of the 'burst edge'. I realise that very short edges carry too little information and on longer edges the burst is diluted. However, unless I have overlooked something, the figure of 0.005 seems to be presented without clear reason: is it a value that is selected to include all short edges? Would increasing the threshold to 0.01, say, add more edges? Was there any kind of optimisation process conducted? To me it seems rather important because if a modest change in threshold increases the number of usable edges, this could add considerably to the analysis.
3. The values for DNAJC11 look unreal. 39:1?? Really? Wow. To me this needs greater exploration. For example, what is the dN/dS ratio for this protein in other species: is the gene generally highly selected or does this represent a one-off event? It is also worth testing whether this represents positive selection for change or 'normal' selection plus a very high mutation rate. By this I mean it could be that any change to the amino acid sequence is beneficial (perhaps!). Alternatively, there could be huge numbers of changes due to a very high mutation rate, and of these a small subset is selected, but because of the high mutation rate, this is still a large number. In reality, I struggle to see how 39:1 can work because it implies that neutral, non-synonymous changes never hitchhike, which seems unlikely. In short, this ratio is so very extreme that I think it needs extra work to show that it is biologically plausible / even possible.

Review form: Reviewer 2 (Edward Braun)**Is the manuscript scientifically sound in its present form?**

Yes

Are the interpretations and conclusions justified by the results?

Yes

Is the language acceptable?

No

Is it clear how to access all supporting data?

No

Do you have any ethical concerns with this paper?

No

Have you any concerns about statistical analyses in this paper?

No

Recommendation?

Major revision is needed (please make suggestions in comments)

Comments to the Author(s)

Overall, I am enthusiastic about the manuscript “Bursts of amino acid replacements in protein evolution” by Stolyarova and colleagues. I have long been interested in the topic of overdispersion in protein evolution and I believe the authors make a substantial advance in the field. I’d like to start my review by highlighting a few general concerns I have regarding the writing, then move on to specific concerns, and finish with a brief discussion of criticisms in the other reviews that do not concern me (i.e., areas where I feel the authors’ work is solid and the criticisms are not valid).

The general criticism that I would articulate about the paper is the rather telegraphic style. Although the material that is there is well-written and clear, there is certainly room to expand the manuscript in a number of places. Royal Society Open Science does not have a word limit. Obviously, the absence of a word limit should not be viewed as a license to include irrelevant information. However, the current manuscript goes much too far in the opposite direction; I believe information that is necessary to understand and evaluate the science has been omitted.

With respect to the writing, I would like to start out with the introduction. The very first paragraph, which is critical to set up the contrast of gradual vs. punctuated evolution seems to be somewhat muddled. As it is currently written, the introduction jumps back and forth in terms of gradualistic and punctuated tempos for evolution and between molecular evolution and whole organism evolution. I think this is what reviewer 3 was responding to in their statement that “...the manuscript attempts to frame [the authors’] research in terms of punctuated evolution and stasis, which [reviewer 3] found much less relevant” than discussing the authors’ work in the context of overdispersion.

Of course, the authors are correct when they responded to this by stating that there “...is no fundamental difference between overdispersed rate of evolution and alternating episodes of evolution and stasis.” However, there can be many causes for alternating episodes of stasis and evolution and the causes of punctuation in molecular and morphological evolution. All of the examples of gradual evolution are molecular, despite the long history of using Brownian motion to model drift in morphological data (see O’Meara et al. 2006). If the authors are going to include examples of morphological evolution they acknowledge that gradual evolution of morphology is possible (indeed, it is often viewed as a null hypothesis).

Another aspect of the introduction that I found troubling is that all of the examples they cite to illustrate gradual evolution are molecular clock studies; to my knowledge molecular clock studies have not tested for overdispersion. Some overdispersed loci might be rejected indirectly when tests of the clock are performed, but modern relaxed clock studies might very well retain some loci that appear non-clocklike due to overdispersion. Studies that use molecular clocks (especially relaxed molecular clocks) are not evidence of gradual evolution per se, unless those studies test for overdispersion (i.e., punctuation) and reject it in favor of gradualism (at least for a subset of loci). Note that there are cases where the overdispersion hypothesis has been explicitly tested and rejected (e.g., Goldman 1994)

This brings us to the causes of punctuated evolution. At the morphological level it seems likely to me that most examples of punctuation reflect selection. This is really what the authors are looking for at the molecular level. However, there are causes of punctuation at the molecular level that reflect neutral evolution rather than selection. The authors acknowledge one of these explanations: gene conversion. However, the authors do not appear to acknowledge that one of

explanation for punctuated molecular evolution offered in the Pagel et al. (2006) study that they cite. Specifically, Pagel et al. (2006) stated that if "...speciation is associated with small founder populations and if genetic isolation is maintained, evolutionary rates can be accelerated at potentially all loci, because the number of loci with alleles governed by drift (neutral plus nearly neutral) is increased." In other words, Pagel et al. (2006) appear to favor a nearly neutral theory explanation for punctuation. Of course, Pagel et al. (2006) do offer selection as an alternative, stating that a "...second general mechanism for divergence is adaptive evolution as species invade new niches." The latter mechanism is the obviously what the authors hope to detect, but it is important that they set up the alternatives carefully.

Overall, I really think the manuscript would be better served with an almost complete rewrite of the introduction. As I said, I agree completely that overdispersion is equivalent to alternating episodes of stasis and evolution. However, I did not feel that the introduction set up the analyses reported in the paper. Moreover, the extreme terseness of the introduction (and the manuscript as a whole) has led me to indicate to the editor that the language in its current form is not acceptable. I debated doing so since I usually state that language is unacceptable if there are statements I simply cannot understand. In this manuscript it is more the case that statements are too terse.

In addition to these problems, I feel that the authors were somewhat sloppy with their citations. I have already brought up the issue that citing studies that use the molecular clock to estimate divergence times as evidence for gradualism is inappropriate; evidence for gradualism requires an actual test for overdispersion. The authors also cited the original Zuckerkandl and Pauling (1965) paper describing molecules as documents of evolutionary history in the context of selectively neutral segments of genomes; Zuckerkandl and Pauling (1965) predated neutral theory (Kimura 1968; King and Jukes 1969). I also fail to see how Cooney et al. (2017) represents an example of "gradual adaptive evolution"; after all, Cooney et al. (2017) state that their results "...support Darwinian⁹ and Simpsonian⁴ ideas of microevolution within adaptive zones and accelerated evolution between distinct adaptive peaks" (numbers in Cooney et al. 2017 quote are citations within that publication). The statement in Cooney et al. (2017) would seem to support a "burst" of substitution where taxa move from one adaptive peak to another. Finally, I was surprised by the failure to cite Gillespie (1991), which I believe to be the classic text describing overdispersed molecular evolution. Finally, the authors should make sure their references are formatted properly; I noticed that Moores et al. (1999) is cited in the text as Ø Moores, Vamosi and Schluter, (1999); the authors should fix this and ensure there are no other mistakes regarding their citations.

Once I got past the introduction I felt the paper was much better and I generally became enthusiastic. The authors present evidence for adaptive overdispersed molecular evolution that I find quite compelling. I only have three issues with the current manuscript; I believe all three to be very important for the authors to address.

1. The issue of gene tree-species tree discordance.

Reviewer 3 also brought up the issue of discordance among gene trees, which the authors dismissed as not being a problem because they "...discarded the genes with poorly reconstructed ancestral sequences, which will be the case for discordant genes (e.g. caused by incomplete lineage sorting)." In other words, the authors reconstructed ancestral sequences using the species tree as the underlying topology and then examined posterior probabilities for the ancestral state reconstructions, discarding loci with low posterior probabilities. I am not at all convinced that poorly reconstructed ancestral sequences (as revealed by low posterior probabilities) are evidence of discordance due to incomplete lineage sorting (ILS).

I will illustrate my assertion that “codeml posterior probabilities for the reconstructed ancestral variant < 0.8 ” will not necessarily highlight loci with discordant gene trees using two scenarios. First, let us imagine a gene encoded by a locus that has a gene tree that is perfectly congruent with the species tree and further postulate that the gene is rapidly evolving. I would expect the high degree of homoplasy that I postulated in this scenario to lead to low codeml posterior probabilities. Second, let us postulate a relatively slowly evolving gene associated with a locus that has a gene tree with some discordance due to ILS. The ILS could lead to some sites in the alignment that appear homoplastic (the appearance of homoplasy given the species tree due to gene tree-species tree discordance is called hemiplasy; see Avise & Robinson 2008) and the ancestral state reconstructions of those sites those might have reduced posterior probabilities. In this example the ancestral state reconstructions may also be incorrect given that they reflect hemiplasy that is misinterpreted as homoplasy, but I do not think it is a given that the reconstructions will definitely have a posterior probability < 0.8 .

Note that this discussion is not a criticism of the chosen posterior probability cut-off; it is an intrinsic problem with the interpretation of ancestral state reconstructions that have low posterior probabilities as reflecting discordance due to ILS. Basically, there are multiple ways for codeml ancestral state reconstruction to have low posterior probabilities for the reconstructed ancestral variants. Discordance due to ILS is likely to be one way since hemiplasy does create the appearance of homoplasy given the species tree. However, unless the authors have done simulations showing that low posterior probabilities in codeml reconstructions are definitely indicative of discordance I do not believe they should state they are.

Does this have a practical impact on the analyses that the authors present? I am uncertain whether it does because I am uncertain how much discordance among gene trees exists in the datasets the authors examined. A simple way for the authors to assess discordance is to look for incongruent gene trees. I am well aware that error in gene tree estimation is a problem for estimating the amount of incongruence among gene trees (for examples of my publications in the area see Patel et al. 2013 and Meiklejohn et al. 2016). However, there is a simple solution for the authors: estimate gene trees with support values using a standard method, such as maximum likelihood (ML) bootstrapping using RAxML (Stamatakis 2014) or IQ-TREE (Nguyen et al. 2015), collapse all branches with less than some amount of support (e.g., 75% standard bootstrap support), and only consider branches with support when examining incongruence. There are important statistical issues with this approach, since there will be some errors (i.e., some branches will appear to be incongruent when they are not). However, it is a better criterion than using posterior probabilities of ancestral state reconstructions because the number of branches with support that conflict with the species tree is a direct measure of incongruence. Ideally, the authors would present this information for all genes (i.e., XX% of the branches in each gene tree received $>75\%$ bootstrap support; YY% of those branches conflicted with the species tree) but they should present this information for the cases where they found significant evidence for bursts.

I actually doubt that this will change their conclusions. It seems likely to me that the bursts define branches that are present in the gene tree. For example, if the 39 non-synonymous substitutions in the amphipod DNAJC11 sequences do not define a clade in the DNAJC11 gene tree then one has to postulate a truly astounding degree of convergence that also happens to appear to be synapomorphic changes when mapped on the species tree. Of course, the DNAJC11 gene tree could be discordant with the species tree elsewhere, but that doesn't matter from the standpoint of inferring bursts of substitutions.

Overall, I am not convinced that gene tree discordance is a problem but the way that the authors address it is not appropriate. At a minimum they should infer ML phylogenies for each gene with a burst and ask whether the branch in the species tree with the burst is present in the gene tree.

There are four possible outcomes for this analysis of the gene tree: 1) the branch with the burst is present in the gene tree and supported by the ML bootstrap; 2) the branch with the burst is present in the gene tree but it is not supported by the ML bootstrap; 3) a branch that conflicts with relevant branch in the species tree is present but that conflicting branch is not supported by the ML bootstrap; and 4) a branch that conflicts with relevant branch in the species tree is present and the conflicting branch is supported by the ML bootstrap. It is only in case #4 that discordance can be said to be a potential problem. The more general analyses of incongruence that I mention above would be interesting, but it is not critical for the authors to prove their point.

2. The puzzle underlying the mutational basis for their extreme cases of overdispersion

I believe that DNAJC11 illustrates something that may be even more interesting than the scenario that the authors postulate. I'd like to start by considering two limits on the rate of accumulation of non-synonymous substitutions given that the authors state that the relevant branch in the species tree has a very short synonymous length ($dS = 0.004$).

Let's begin with what I'll call the "neutral limit." There are there are 1515 sites in the alignment (I am using gap free sites, but the total number of sites is not much larger). If we assume the number of synonymous sites is $\sim 1/3$ of the total number of the expected number of synonymous substitutions for that branch is ~ 2 . If we assume synonymous sites are selectively neutral the rate of accumulation for synonymous substitutions is the mutation rate (Kimura 1968). Obviously, many non-synonymous sites will be subject to purifying selection but this dS would imply that, if all non-synonymous sites were selectively neutral (i.e., if $dN/dS = 1.0$), that the expected number of non-synonymous substitutions is only ~ 4 . However, the authors observed 39 non-synonymous substitutions in this locus; the probability of this many non-synonymous substitutions when the expected number of substitutions if we assume a Poisson process is extremely low. Of course, the authors would reply "that is our point" to this description of the "neutral limit." However, my point in articulating this to place a lower limit on the number of non-synonymous substitutions that it is reasonable to expect if selection for changes at individual sites is relatively weak; the expectation given neutrality provides that lower limit. I would add that this "neutral limit" is really just a restatement of the authors' null hypothesis; in fact, they stated that they "...assumed the neutral null model ($dN=dS$)" and emphasized that this will make "...it possible to detect only the longest and the fastest[sic] bursts."

Another way of articulating the fact that the authors' method can "...detect only the longest and the fastest[sic] bursts" is to say that the non-synonymous substitutions must be subject to strong selection. However, this need for strong selection on new non-synonymous variants brings us to a different limit that I'll call the "Hill-Robertson limit." If we assume the selection for individual non-synonymous changes is strong then those alleles should be fixed rapidly. This would tend to eliminate other alleles, even if those other alleles are advantageous relative to the ancestral allele. This is just a restatement of the well-known Hill & Robertson (1966) effect. Given Hill-Robertson interference one must assume that a large number of potential non-synonymous changes become advantageous and then occur successively. In other words, Hill-Robertson interference requires one to postulate the following scenario: 1) a new advantageous mutation enters the population; 2) the new mutation becomes fixed in the (metaphorical) blink of an eye; 3) another advantageous mutation enters the population (in the background of newly fixed advantageous allele); and 4) this cycle repeats for a total of 39 times. Moreover, even if one imagines an environmental shift occurs that favors specific substitutions at >30 sites one would think that epistatic interactions among sites within the protein would create issues (though I suppose one could also argue that epistasis might create new "opportunities" for changes that are subject to positive selection). Perhaps there are parts of parameter space (i.e., mutation rate, generation time, effective population size, selection coefficients on individual sites, etc.) where this can occur. But it seems to me that the "good" part of parameter space might be very small.

I can see two ways out of the narrow space created by these two limits: 1) recombination rates are high enough to overcome Hill-Robertson interference; or 2) the mutation that generated these non-synonymous changes altered multiple sites. The first explanation would probably require that there is a reasonable rate of recombination between sites that are ~10 bp apart (some of the amino acid substitutions uniting the group defined by shading in Fig. 1 are 2-3 amino acids apart). The second might imply that only a subset of the non-synonymous substitutions are advantageous; the remainder were simply “dragged” to fixation by the advantageous change(s). Both possibilities are interesting. It is debatable whether this argument applies to all of their examples, but the authors were looking for exceptional outcomes. Perhaps the exceptional outcomes indicate that surprising molecular mechanisms exist.

Finally, the authors should fix the misspelling of “fastest” indicated above using “[sic]”.

3. The base composition of synonymous sites

dS can be underestimated when there is strong codon bias (Rabinowicz et al. 1999). The authors should present the base composition (and variation among loci in their base composition) for the data used to calculate dS.

4. The location of non-synonymous substitutions

The secondary structure of the Catarrhini PKR sequences is shown but the secondary structure of the DNAJC11 sequences is not shown. I'm not convinced that presenting the secondary structure is important unless other information is provided, but the authors should be consistent. It is more important to indicate the locations of the substitutions relative to the boundaries of protein domains, which the authors do for PKR. It would be good to do this for DNAJC11 as well; for example, how many of the 39 substitutions are located in the J domain?

Finally, there are two words that should be fixed in the Fig. 3 legend: “others” and “in”. I have indicated them in CAPS below:

“Fig. 3. Alignment of PKR genes of Catarrhini (fragments) containing 18 nonsynonymous substitutions on the internal edge ancestral to *Macaca mulatta* and *Macaca fascicularis*. The majority of substitutions (14) occurred in the kinase domain of the protein, the OTHERS fall IN the dsRNA binding motif (DRBM1). Alleles derived in the adaptive burst are shown in bold.”

5. Data accessibility

I was unable to access the Dryad data package. Assuming that it is made available and includes (at a minimum) the following:

- 1) Nucleotide sequence alignments of the loci they examined (ideally all sequences) in a readily usable format (e.g., nexus or relaxed phylip).
- 2) The phylogenetic trees they used for their analyses in newick format.
- 3) The gene names and functional annotations (especially the blast2GO data for the amphipods).

I think it will be appropriate.

I would like to emphasize, despite the fact that I wrote a long review pointing out issues, that I am enthusiastic about this manuscript. I think the authors have presented good evidence for their central assertion that there were adaptive bursts occur in small numbers of proteins. Thus, I

believe the science is sound. I also think their conclusions are of broad interest. I hope the authors and editor find these comments helpful.

Edward L. Braun
Professor of Biology
University of Florida
Gainesville, FL 32611

REFERENCES:

Avise JC, Robinson TJ. 2008. Hemiplasy: a new term in the lexicon of phylogenetics. *Systematic Biology*, 57(3): 503-507. DOI: 10.1080/10635150802164587

Gillespie JH. 1991. *The Causes of Molecular Evolution* (Oxford Series in Ecology and Evolution, edited by May RM and Harvey PH). Oxford University Press, Oxford. ISBN-10: 0195092716

Goldman N. 1994. Variance to mean ratio, $R(t)$, for Poisson processes on phylogenetic trees. *Molecular Phylogenetics and Evolution*, 3(3): 230-239 DOI: 10.1006/mpev.1994.1025

Hill WG, Robertson A. 1966. The effect of linkage on limits to artificial selection. *Genetical Research*, 8: 269-294 DOI: 10.1017/S0016672300010156

Kimura M. 1968. Evolutionary rate at the molecular level. *Nature* 217: 624-626
DOI:10.1038/217624a0

King JL, Jukes TH. 1969. Non-darwinian evolution. *Science*, 164(3881): 788-798 DOI: 10.1126/science.164.3881.788

Meiklejohn KA, Faircloth BC, Glenn TC, Kimball RT, Braun EL. 2016. Analysis of a rapid evolutionary radiation using ultraconserved elements: evidence for a bias in some multispecies coalescent methods. *Systematic Biology*, 65(4): 612-627. DOI: 10.1093/sysbio/syw014

Nguyen L-T, Schmidt HA, von Haeseler A, Minh BQ. 2015. IQ-TREE: A fast and effective stochastic algorithm for estimating maximum likelihood phylogenies. *Molecular Biology and Evolution*, 32: 268-274 DOI: 10.1093/molbev/msu300

O'Meara BC, Ané C, Sanderson MJ, Wainwright PC. 2006. Testing for different rates of continuous trait evolution using likelihood. *Evolution*, 60(5): 922-933. DOI: 10.1111/j.0014-3820.2006.tb01171.x

Pagel M, Venditti C, Meade A. 2006. Large punctuational contribution of speciation to evolutionary divergence at the molecular level. *Science*, 314(5796): 119-21. DOI: 10.1126/science.1129647 Erratum in: *Science*, 2006 314: 925.

Patel S, Kimball RT, Braun EL. 2013. Error in phylogenetic estimation for bushes in the tree of life. *Journal of Phylogenetics and Evolutionary Biology* 1: 110 DOI: 10.4172/2329-9002.1000110

Rabinowicz PD, Braun EL, Wolfe AD, Bowen B, Grotewold E. 1999. Maize R2R3 Myb genes: sequence analysis reveals amplification in the higher plants. *Genetics*, 153(1): 427-444

Stamatakis A. 2014. RAxML version 8: a tool for phylogenetic analysis and post-analysis of large phylogenies. *Bioinformatics*, 30(9): 1312-1313 DOI: 10.1093/bioinformatics/btu033

Decision letter (RSOS-181095.R0)

18-Oct-2018

Dear Ms Stolyarova,

The editors assigned to your paper ("Bursts of amino acid replacements in protein evolution") have now received comments from reviewers.

Both reviewers are positive about the publication of your paper. But both reviewers raise a number of points, some of them substantive, that would merit further consideration, amplification in some cases and revision of your manuscript. We would like you to revise your paper in accordance with the referee's suggestions which can be found below (not including confidential reports to the Editor). Please note this decision does not guarantee eventual acceptance.

Please submit a copy of your revised paper before 10-Nov-2018. Please note that the revision deadline will expire at 00.00am on this date. If we do not hear from you within this time then it will be assumed that the paper has been withdrawn. In exceptional circumstances, extensions may be possible if agreed with the Editorial Office in advance. We do not allow multiple rounds of revision so we urge you to make every effort to fully address all of the comments at this stage. If deemed necessary by the Editors, your manuscript will be sent back to one or more of the original reviewers for assessment. If the original reviewers are not available, we may invite new reviewers.

- Data accessibility

It is a condition of publication that all supporting data are made available either as supplementary information or preferably in a suitable permanent repository. The data accessibility section should state where the article's supporting data can be accessed. This section should also include details, where possible of where to access other relevant research materials such as statistical tools, protocols, software etc can be accessed. If the data have been deposited in an external repository this section should list the database, accession number and link to the DOI

for all data from the article that have been made publicly available. Data sets that have been deposited in an external repository and have a DOI should also be appropriately cited in the manuscript and included in the reference list.

If you wish to submit your supporting data or code to Dryad (<http://datadryad.org/>), or modify your current submission to dryad, please use the following link:
<http://datadryad.org/submit?journalID=RSOS&manu=RSOS-181095>

- **Competing interests**

- **Authors' contributions**

- **Acknowledgements**

- **Funding statement**

Please note that Royal Society Open Science charge article processing charges for all new submissions that are accepted for publication. Charges will also apply to papers transferred to Royal Society Open Science from other Royal Society Publishing journals, as well as papers submitted as part of our collaboration with the Royal Society of Chemistry (<http://rsos.royalsocietypublishing.org/chemistry>). If your manuscript is newly submitted and subsequently accepted for publication, you will be asked to pay the article processing charge, unless you request a waiver and this is approved by Royal Society Publishing. You can find out more about the charges at <http://rsos.royalsocietypublishing.org/page/charges>. Should you have any queries, please contact openscience@royalsociety.org.

Kind regards,
Royal Society Open Science Editorial Office
Royal Society Open Science

on behalf of Dr Steve Brown (Subject Editor)
openscience@royalsociety.org

Comments to Author:

Reviewers' Comments to Author:
Reviewer: 1

Comments to the Author(s)

I rather like this paper though for me it tantalises more than it reveals. I feel it could be fleshed out a bit more, particularly with regard to the biggest 'hit' in point 3 below:

1. There is an assertion that these are the only sets of genomes that are suitable. This may be so, but it would be nice to hear the reasons why obvious candidates like *Drosophila* are not included.
2. I would like to see more justification for the size of the 'burst edge'. I realise that very short edges carry too little information and on longer edges the burst is diluted. However, unless I have overlooked something, the figure of 0.005 seems to be presented without clear reason: is it a value that is selected to include all short edges? Would increasing the threshold to 0.01, say, add more edges? Was there any kind of optimisation process conducted? To me it seems rather important because if a modest change in threshold increases the number of usable edges, this could add considerably to the analysis.
3. The values for DNAJC11 look unreal. 39:1?? Really? Wow. To me this needs greater exploration. For example, what is the dN/dS ratio for this protein in other species: is the gene generally highly selected or does this represent a one-off event? It is also worth testing whether this represents positive selection for change or 'normal' selection plus a very high mutation rate. By this I mean it could be that any change to the amino acid sequence is beneficial (perhaps!). Alternatively, there could be huge numbers of changes due to a very high mutation rate, and of these a small subset is selected, but because of the high mutation rate, this is still a large number. In reality, I struggle to see how 39:1 can work because it implies that neutral, non-synonymous changes never hitchhike, which seems unlikely. In short, this ratio is so very extreme that I think it needs extra work to show that it is biologically plausible / even possible.

Reviewer: 2

Comments to the Author(s)

Overall, I am enthusiastic about the manuscript "Bursts of amino acid replacements in protein evolution" by Stolyarova and colleagues. I have long been interested in the topic of overdispersion in protein evolution and I believe the authors make a substantial advance in the field. I'd like to start my review by highlighting a few general concerns I have regarding the writing, then move on to specific concerns, and finish with a brief discussion of criticisms in the other reviews that do not concern me (i.e., areas where I feel the authors' work is solid and the criticisms are not valid).

The general criticism that I would articulate about the paper is the rather telegraphic style. Although the material that is there is well-written and clear, there is certainly room to expand the manuscript in a number of places. Royal Society Open Science does not have a word limit. Obviously, the absence of a word limit should not be viewed as a license to include irrelevant information. However, the current manuscript goes much too far in the opposite direction; I believe information that is necessary to understand and evaluate the science has been omitted.

With respect to the writing, I would like to start out with the introduction. The very first paragraph, which is critical to set up the contrast of gradual vs. punctuated evolution seems to be somewhat muddled. As it is currently written, the introduction jumps back and forth in terms of gradualistic and punctuated tempos for evolution and between molecular evolution and whole organism evolution. I think this is what reviewer 3 was responding to in their statement that "...the manuscript attempts to frame [the authors'] research in terms of punctuated evolution and stasis, which [reviewer 3] found much less relevant" than discussing the authors' work in the context of overdispersion.

Of course, the authors are correct when they responded to this by stating that there "...is no fundamental difference between overdispersed rate of evolution and alternating episodes of evolution and stasis." However, there can be many causes for alternating episodes of stasis and evolution and the causes of punctuation in molecular and morphological evolution. All of the examples of gradual evolution are molecular, despite the long history of using Brownian motion to model drift in morphological data (see O'Meara et al. 2006). If the authors are going to include examples of morphological evolution they acknowledge that gradual evolution of morphology is possible (indeed, it is often viewed as a null hypothesis).

Another aspect of the introduction that I found troubling is that all of the examples they cite to illustrate gradual evolution are molecular clock studies; to my knowledge molecular clock studies have not tested for overdispersion. Some overdispersed loci might be rejected indirectly when tests of the clock are performed, but modern relaxed clock studies might very well retain some loci that appear non-clocklike due to overdispersion. Studies that use molecular clocks (especially relaxed molecular clocks) are not evidence of gradual evolution per se, unless those studies test for overdispersion (i.e., punctuation) and reject it in favor of gradualism (at least for a subset of loci). Note that there are cases where the overdispersion hypothesis has been explicitly tested and rejected (e.g., Goldman 1994)

This brings us to the causes of punctuated evolution. At the morphological level it seems likely to me that most examples of punctuation reflect selection. This is really what the authors are looking for at the molecular level. However, there are causes of punctuation at the molecular level that reflect neutral evolution rather than selection. The authors acknowledge one of these explanations: gene conversion. However, the authors do not appear to acknowledge that one of explanation for punctuated molecular evolution offered in the Pagel et al. (2006) study that they cite. Specifically, Pagel et al. (2006) stated that "...speciation is associated with small founder populations and if genetic isolation is maintained, evolutionary rates can be accelerated at potentially all loci, because the number of loci with alleles governed by drift (neutral plus nearly neutral) is increased." In other words, Pagel et al. (2006) appear to favor a nearly neutral theory explanation for punctuation. Of course, Pagel et al. (2006) do offer selection as an alternative, stating that a "...second general mechanism for divergence is adaptive evolution as species invade new niches." The latter mechanism is the obviously what the authors hope to detect, but it is important that they set up the alternatives carefully.

Overall, I really think the manuscript would be better served with an almost complete rewrite of the introduction. As I said, I agree completely that overdispersion is equivalent to alternating episodes of stasis and evolution. However, I did not feel that the introduction set up the analyses reported in the paper. Moreover, the extreme terseness of the introduction (and the manuscript as a whole) has led me to indicate to the editor that the language in its current form is not acceptable. I debated doing so since I usually state that language is unacceptable if there are statements I simply cannot understand. In this manuscript it is more the case that statements are too terse.

In addition to these problems, I feel that the authors were somewhat sloppy with their citations. I have already brought up the issue that citing studies that use the molecular clock to estimate divergence times as evidence for gradualism is inappropriate; evidence for gradualism requires an actual test for overdispersion. The authors also cited the original Zuckerkandl and Pauling (1965) paper describing molecules as documents of evolutionary history in the context of selectively neutral segments of genomes; Zuckerkandl and Pauling (1965) predated neutral theory (Kimura 1968; King and Jukes 1969). I also fail to see how Cooney et al. (2017) represents an example of “gradual adaptive evolution”; after all, Cooney et al. (2017) state that their results “...support Darwinian⁹ and Simpsonian⁴ ideas of microevolution within adaptive zones and accelerated evolution between distinct adaptive peaks” (numbers in Cooney et al. 2017 quote are citations within that publication). The statement in Cooney et al. (2017) would seem to support a “burst” of substitution where taxa move from one adaptive peak to another. Finally, I was surprised by the failure to cite Gillespie (1991), which I believe to be the classic text describing overdispersed molecular evolution. Finally, the authors should make sure their references are formatted properly; I noticed that Moores et al. (1999) is cited in the text as Ø Moores, Vamosi and Schluter, (1999); the authors should fix this and ensure there are no other mistakes regarding their citations.

Once I got past the introduction I felt the paper was much better and I generally became enthusiastic. The authors present evidence for adaptive overdispersed molecular evolution that I find quite compelling. I only have three issues with the current manuscript; I believe all three to be very important for the authors to address.

1. The issue of gene tree-species tree discordance.

Reviewer 3 also brought up the issue of discordance among gene trees, which the authors dismissed as not being a problem because they “...discarded the genes with poorly reconstructed ancestral sequences, which will be the case for discordant genes (e.g. caused by incomplete lineage sorting).” In other words, the authors reconstructed ancestral sequences using the species tree as the underlying topology and then examined posterior probabilities for the ancestral state reconstructions, discarding loci with low posterior probabilities. I am not at all convinced that poorly reconstructed ancestral sequences (as revealed by low posterior probabilities) are evidence of discordance due to incomplete lineage sorting (ILS).

I will illustrate my assertion that “codeml posterior probabilities for the reconstructed ancestral variant < 0.8 ” will not necessarily highlight loci with discordant gene trees using two scenarios. First, let us imagine a gene encoded by a locus that has a gene tree that is perfectly congruent with the species tree and further postulate that the gene is rapidly evolving. I would expect the high degree of homoplasy that I postulated in this scenario to lead to low codeml posterior probabilities. Second, let us postulate a relatively slowly evolving gene associated with a locus that has a gene tree with some discordance due to ILS. The ILS could lead to some sites in the alignment that appear homoplastic (the appearance of homoplasy given the species tree due to gene tree-species tree discordance is called hemiplasy; see Avise & Robinson 2008) and the ancestral state reconstructions of those sites those might have reduced posterior probabilities. In this example the ancestral state reconstructions may also be incorrect given that they reflect hemiplasy that is misinterpreted as homoplasy, but I do not think it is a given that the reconstructions will definitely have a posterior probability < 0.8 .

Note that this discussion is not a criticism of the chosen posterior probability cut-off; it is an intrinsic problem with the interpretation of ancestral state reconstructions that have low posterior probabilities as reflecting discordance due to ILS. Basically, there are multiple ways for codeml ancestral state reconstruction to have low posterior probabilities for the reconstructed ancestral

variants. Discordance due to ILS is likely to be one way since hemiplasy does create the appearance of homoplasy given the species tree. However, unless the authors have done simulations showing that low posterior probabilities in codeml reconstructions are definitely indicative of discordance I do not believe they should state they are.

Does this have a practical impact on the analyses that the authors present? I am uncertain whether it does because I am uncertain how much discordance among gene trees exists in the datasets the authors examined. A simple way for the authors to assess discordance is to look for incongruent gene trees. I am well aware that error in gene tree estimation is a problem for estimating the amount of incongruence among gene trees (for examples of my publications in the area see Patel et al. 2013 and Meiklejohn et al. 2016). However, there is a simple solution for the authors: estimate gene trees with support values using a standard method, such as maximum likelihood (ML) bootstrapping using RAxML (Stamatakis 2014) or IQ-TREE (Nguyen et al. 2015), collapse all branches with less than some amount of support (e.g., 75% standard bootstrap support), and only consider branches with support when examining incongruence. There are important statistical issues with this approach, since there will be some errors (i.e., some branches will appear to be incongruent when they are not). However, it is a better criterion than using posterior probabilities of ancestral state reconstructions because the number of branches with support that conflict with the species tree is a direct measure of incongruence. Ideally, the authors would present this information for all genes (i.e., XX% of the branches in each gene tree received >75% bootstrap support; YY% of those branches conflicted with the species tree) but they should present this information for the cases where they found significant evidence for bursts.

I actually doubt that this will change their conclusions. It seems likely to me that the bursts define branches that are present in the gene tree. For example, if the 39 non-synonymous substitutions in the amphipod DNAJC11 sequences do not define a clade in the DNAJC11 gene tree then one has to postulate a truly astounding degree of convergence that also happens to appear to be synapomorphic changes when mapped on the species tree. Of course, the DNAJC11 gene tree could be discordant with the species tree elsewhere, but that doesn't matter from the standpoint of inferring bursts of substitutions.

Overall, I am not convinced that gene tree discordance is a problem but the way that the authors address it is not appropriate. At a minimum they should infer ML phylogenies for each gene with a burst and ask whether the branch in the species tree with the burst is present in the gene tree. There are four possible outcomes for this analysis of the gene tree: 1) the branch with the burst is present in the gene tree and supported by the ML bootstrap; 2) the branch with the burst is present in the gene tree but it is not supported by the ML bootstrap; 3) a branch that conflicts with relevant branch in the species tree is present but that conflicting branch is not supported by the ML bootstrap; and 4) a branch that conflicts with relevant branch in the species tree is present and the conflicting branch is supported by the ML bootstrap. It is only in case #4 that discordance can be said to be a potential problem. The more general analyses of incongruence that I mention above would be interesting, but it is not critical for the authors to prove their point.

2. The puzzle underlying the mutational basis for their extreme cases of overdispersion

I believe that DNAJC11 illustrates something that may be even more interesting than the scenario that the authors postulate. I'd like to start by considering two limits on the rate of accumulation of non-synonymous substitutions given that the authors state that the relevant branch in the species tree has a very short synonymous length ($dS = 0.004$).

Let's begin with what I'll call the "neutral limit." There are there are 1515 sites in the alignment (I am using gap free sites, but the total number of sites is not much larger). If we assume the number of synonymous sites is $\sim 1/3$ of the total number of the expected number of synonymous

substitutions for that branch is ~ 2 . If we assume synonymous sites are selectively neutral the rate of accumulation for synonymous substitutions is the mutation rate (Kimura 1968). Obviously, many non-synonymous sites will be subject to purifying selection but this dS would imply that, if all non-synonymous sites were selectively neutral (i.e., if $dN/dS = 1.0$), that the expected number of non-synonymous substitutions is only ~ 4 . However, the authors observed 39 non-synonymous substitutions in this locus; the probability of this many non-synonymous substitutions when the expected number of substitutions if we assume a Poisson process is extremely low. Of course, the authors would reply “that is our point” to this description of the “neutral limit.” However, my point in articulating this to place a lower limit on the number of non-synonymous substitutions that it is reasonable to expect if selection for changes at individual sites is relatively weak; the expectation given neutrality provides that lower limit. I would add that this “neutral limit” is really just a restatement of the authors’ null hypothesis; in fact, they stated that they “...assumed the neutral null model ($dN=dS$)” and emphasized that this will make “...it possible to detect only the longest and the fastest[sic] bursts.”

Another way of articulating the fact that the authors’ method can “...detect only the longest and the fastest[sic] bursts” is to say that the non-synonymous substitutions must be subject to strong selection. However, this need for strong selection on new non-synonymous variants brings us to a different limit that I’ll call the “Hill-Robertson limit.” If we assume the selection for individual non-synonymous changes is strong then those alleles should be fixed rapidly. This would tend to eliminate other alleles, even if those other alleles are advantageous relative to the ancestral allele. This is just a restatement of the well-known Hill & Robertson (1966) effect. Given Hill-Robertson interference one must assume that a large number of potential non-synonymous changes become advantageous and then occur successively. In other words, Hill-Robertson interference requires one to postulate the following scenario: 1) a new advantageous mutation enters the population; 2) the new mutation becomes fixed in the (metaphorical) blink of an eye; 3) another advantageous mutation enters the population (in the background of newly fixed advantageous allele); and 4) this cycle repeats for a total of 39 times. Moreover, even if one imagines an environmental shift occurs that favors specific substitutions at >30 sites one would think that epistatic interactions among sites within the protein would create issues (though I suppose one could also argue that epistasis might create new “opportunities” for changes that are subject to positive selection). Perhaps there are parts of parameter space (i.e., mutation rate, generation time, effective population size, selection coefficients on individual sites, etc.) where this can occur. But it seems to me that the “good” part of parameter space might be very small.

I can see two ways out of the narrow space created by these two limits: 1) recombination rates are high enough to overcome Hill-Robertson interference; or 2) the mutation that generated these non-synonymous changes altered multiple sites. The first explanation would probably require that there is a reasonable rate of recombination between sites that are ~ 10 bp apart (some of the amino acid substitutions uniting the group defined by shading in Fig. 1 are 2-3 amino acids apart). The second might imply that only a subset of the non-synonymous substitutions are advantageous; the remainder were simply “dragged” to fixation by the advantageous change(s). Both possibilities are interesting. It is debatable whether this argument applies to all of their examples, but the authors were looking for exceptional outcomes. Perhaps the exceptional outcomes indicate that surprising molecular mechanisms exist.

Finally, the authors should fix the misspelling of “fastest” indicated above using “[sic]”.

3. The base composition of synonymous sites

dS can be underestimated when there is strong codon bias (Rabinowicz et al. 1999). The authors should present the base composition (and variation among loci in their base composition) for the data used to calculate dS .

4. The location of non-synonymous substitutions

The secondary structure of the Catarrhini PKR sequences is shown but the secondary structure of the DNAJC11 sequences is not shown. I'm not convinced that presenting the secondary structure is important unless other information is provided, but the authors should be consistent. It is more important to indicate the locations of the substitutions relative to the boundaries of protein domains, which the authors do for PKR. It would be good to do this for DNAJC11 as well; for example, how many of the 39 substitutions are located in the J domain?

Finally, there are two words that should be fixed in the Fig. 3 legend: "others" and "in". I have indicated them in CAPS below:

"Fig. 3. Alignment of PKR genes of Catarrhini (fragments) containing 18 nonsynonymous substitutions on the internal edge ancestral to *Macaca mulatta* and *Macaca fascicularis*. The majority of substitutions (14) occurred in the kinase domain of the protein, the OTHERS fall IN the dsRNA binding motif (DRBM1). Alleles derived in the adaptive burst are shown in bold."

5. Data accessibility

I was unable to access the Dryad data package. Assuming that it is made available and includes (at a minimum) the following:

- 1) Nucleotide sequence alignments of the loci they examined (ideally all sequences) in a readily usable format (e.g., nexus or relaxed phylip).
- 2) The phylogenetic trees they used for their analyses in newick format.
- 3) The gene names and functional annotations (especially the blast2GO data for the amphipods).

I think it will be appropriate.

I would like to emphasize, despite the fact that I wrote a long review pointing out issues, that I am enthusiastic about this manuscript. I think the authors have presented good evidence for their central assertion that there were adaptive bursts occur in small numbers of proteins. Thus, I believe the science is sound. I also think their conclusions are of broad interest.

I hope the authors and editor find these comments helpful.

Edward L. Braun
Professor of Biology
University of Florida
Gainesville, FL 32611

REFERENCES:

- Avice JC, Robinson TJ. 2008. Hemiplasy: a new term in the lexicon of phylogenetics. *Systematic Biology*, 57(3): 503-507. DOI: 10.1080/10635150802164587
- Gillespie JH. 1991. *The Causes of Molecular Evolution* (Oxford Series in Ecology and Evolution, edited by May RM and Harvey PH). Oxford University Press, Oxford. ISBN-10: 0195092716
- Goldman N. 1994. Variance to mean ratio, $R(t)$, for Poisson processes on phylogenetic trees. *Molecular Phylogenetics and Evolution*, 3(3): 230-239 DOI: 10.1006/mpev.1994.1025

Hill WG, Robertson A. 1966. The effect of linkage on limits to artificial selection. *Genetical Research*, 8: 269–294 DOI: 10.1017/S0016672300010156

Kimura M. 1968. Evolutionary rate at the molecular level. *Nature* 217: 624-626
DOI:10.1038/217624a0

King JL, Jukes TH. 1969. Non-darwinian evolution. *Science*, 164(3881): 788-798 DOI: 10.1126/science.164.3881.788

Meiklejohn KA, Faircloth BC, Glenn TC, Kimball RT, Braun EL. 2016. Analysis of a rapid evolutionary radiation using ultraconserved elements: evidence for a bias in some multispecies coalescent methods. *Systematic Biology*, 65(4): 612-627. DOI: 10.1093/sysbio/syw014

Nguyen L-T, Schmidt HA, von Haeseler A, Minh BQ. 2015. IQ-TREE: A fast and effective stochastic algorithm for estimating maximum likelihood phylogenies. *Molecular Biology and Evolution*, 32: 268-274 DOI: 10.1093/molbev/msu300

O’Meara BC, Ané C, Sanderson MJ, Wainwright PC. 2006. Testing for different rates of continuous trait evolution using likelihood. *Evolution*, 60(5): 922-933. DOI: 10.1111/j.0014-3820.2006.tb01171.x

Pagel M, Venditti C, Meade A. 2006. Large punctuational contribution of speciation to evolutionary divergence at the molecular level. *Science*, 314(5796): 119-21. DOI: 10.1126/science.1129647 Erratum in: *Science*, 2006 314: 925.

Patel S, Kimball RT, Braun EL. 2013. Error in phylogenetic estimation for bushes in the tree of life. *Journal of Phylogenetics and Evolutionary Biology* 1: 110 DOI: 10.4172/2329-9002.1000110

Rabinowicz PD, Braun EL, Wolfe AD, Bowen B, Grotewold E. 1999. Maize R2R3 Myb genes: sequence analysis reveals amplification in the higher plants. *Genetics*, 153(1): 427-444

Stamatakis A. 2014. RAxML version 8: a tool for phylogenetic analysis and post-analysis of large phylogenies. *Bioinformatics*, 30(9): 1312-1313 DOI: 10.1093/bioinformatics/btu

Author's Response to Decision Letter for (RSOS-181095.R0)

See Appendix A.

RSOS-181095.R1 (Revision)

Review form: Reviewer 1 (William Amos)

Is the manuscript scientifically sound in its present form?

Yes

Are the interpretations and conclusions justified by the results?

Yes

Is the language acceptable?

Yes

Is it clear how to access all supporting data?

Yes

Do you have any ethical concerns with this paper?

No

Have you any concerns about statistical analyses in this paper?

No

Recommendation?

Accept with minor revision (please list in comments)

Comments to the Author(s)

I feel my concerns have largely been addressed. However, I still worry about the 'monstrous hit'. This is such an extreme case that it needs to be treated with great caution, particularly since it is mitochondrial-associated, which raises the possibility of contamination from homologous transcripts from related genes (nuclear vs mtDNA copies / paternally inherited mtDNA etc.). I realise that there has been filtering by dS value but wonder whether this is enough. Mitochondrial sequences have highly polarised based frequencies which could greatly reduce dS values relative to a naive JK model. For me, more analysis is needed, for example by including a tree based synonymous substitutions for the gene that will hopefully be concordant with the overall tree, yet diverge strongly from an equivalent tree based on non-synonymous changes. Extreme observations require extreme support before they can be taken seriously. This is particularly true when the extreme observation is one of very few similar observations.

Decision letter (RSOS-181095.R1)

07-Feb-2019

Dear Ms Stolyarova:

On behalf of the Editors, I am pleased to inform you that your Manuscript RSOS-181095.R1 entitled "Bursts of amino acid replacements in protein evolution" has been accepted for publication in Royal Society Open Science subject to minor revision in accordance with the referee suggestions. Please find the referee's comments at the end of this email.

The reviewer and Subject Editor have recommended publication, but we would like you to take into account the comments of the reviewer of your manuscript, which may require some final minor modifications of your paper. Therefore, I invite you to respond to the comments and revise your manuscript.

- Ethics statement

If your study uses humans or animals please include details of the ethical approval received, including the name of the committee that granted approval. For human studies please also detail

whether informed consent was obtained. For field studies on animals please include details of all permissions, licences and/or approvals granted to carry out the fieldwork.

- Data accessibility

If you wish to submit your supporting data or code to Dryad (<http://datadryad.org/>), or modify your current submission to dryad, please use the following link:
<http://datadryad.org/submit?journalID=RSOS&manu=RSOS-181095.R1>

- Competing interests

- Authors' contributions

- Acknowledgements

- Funding statement

Because the schedule for publication is very tight, it is a condition of publication that you submit the revised version of your manuscript before 16-Feb-2019. Please note that the revision deadline

will expire at 00.00am on this date. If you do not think you will be able to meet this date please let me know immediately.

on behalf of Dr Steve Brown (Subject Editor)
openscience@royalsociety.org

Associate Editor Comments to Author (Dr Steve Brown):

Thank you for the many detailed responses, which will now be considered by the reviewers of your paper.

Reviewer comments to Author:

Reviewer: 1

Comments to the Author(s)

I feel my concerns have largely been addressed. However, I still worry about the 'monstrous hit'. This is such an extreme case that it needs to be treated with great caution, particularly since it is mitochondrial-associated, which raises the possibility of contamination from homologous transcripts from related genes (nuclear vs mtDNA copies / paternally inherited mtDNA etc.). I realise that there has been filtering by dS value but wonder whether this is enough.

Mitochondrial sequences have highly polarised based frequencies which could greatly reduce dS values relative to a naive JK model. For me, more analysis is needed, for example by including a tree based synonymous substitutions for the gene that will hopefully be concordant with the overall tree, yet diverge strongly from an equivalent tree based on non-synonymous changes. Extreme observations require extreme support before they can be taken seriously. This is particularly true when the extreme observation is one of very few similar observations.

Author's Response to Decision Letter for (RSOS-181095.R1)

See Appendix B.

Decision letter (RSOS-181095.R2)

25-Feb-2019

Dear Ms Stolyarova,

I am pleased to inform you that your manuscript entitled "Bursts of amino acid replacements in protein evolution" is now accepted for publication in Royal Society Open Science.

on behalf of Professor Steve Brown (Subject Editor)
openscience@royalsociety.org

Appendix A

Dear Dr Steve Brown,

Thank you for the opportunity to revise our work. Please consider the revised manuscript "Bursts of amino acid replacements in protein evolution" for publication in Royal Society Open Science (Manuscript ID RSOS-181095). We carefully reviewed the comments from the referees and have made the corresponding changes in the manuscript. Our point-by-point response to reviewers' comments is given below (in bold). We hope you find the revised manuscript acceptable for publication.

Best regards,
Anastasia Stolyarova
(on behalf of all authors)

=====

18-Oct-2018

Dear Ms Stolyarova,

The editors assigned to your paper ("Bursts of amino acid replacements in protein evolution") have now received comments from reviewers.

Both reviewers are positive about the publication of your paper. But both reviewers raise a number of points, some of them substantive, that would merit further consideration, amplification in some cases and revision of your manuscript. We would like you to revise your paper in accordance with the referee's suggestions which can be found below (not including confidential reports to the Editor). Please note this decision does not guarantee eventual acceptance.

<...>

on behalf of Dr Steve Brown (Subject Editor)
openscience@royalsociety.org

Comments to Author:

Reviewers' Comments to Author:

Reviewer: 1

Comments to the Author(s)

I rather like this paper though for me it tantalises more than it reveals. I feel it could be fleshed out a bit more, particularly with regard to the biggest 'hit' in point 3 below:

1. There is an assertion that these are the only sets of genomes that are suitable. This may be so, but it would be nice to hear the reasons why obvious candidates like *Drosophila* are not included.

We thank the reviewer for their careful reading of our manuscript. In *Drosophila* phylogeny, as it is known now, the shortest internal edge is ~10 times longer than our cut-off value of 0.005. The edges of such length do not allow to detect short bursts of nonsynonymous substitutions, because, as the reviewer fairly notices below, the bursts become “diluted” and hard to detect.

2. I would like to see more justification for the size of the ‘burst edge’. I realise that very short edges carry too little information and on longer edges the burst is diluted. However, unless I have overlooked something, the figure of 0.005 seems to be presented without clear reason: is it a value that is selected to include all short edges? Would increasing the threshold to 0.01, say, add more edges? Was there any kind of optimisation process conducted? To me it seems rather important because if a modest change in threshold increases the number of usable edges, this could add considerably to the analysis.

Yes - we considered all edges with lengths (in the units of dS) below or equal to 0.005. Changing the threshold would of course increase the number of edges, but not necessary the number of detected bursts. Longer edges would require a larger expected number of substitutions used in our null model, so it would be harder to identify significant excesses of nonsynonymous substitutions on these edges. Moreover, we should be aware of multiple testing correction, so we didn’t want to extend the number of used edges without need. Besides that, none of the confirmed bursts is spread to the adjacent edges, so we don’t have any evidence that the characteristic duration of a burst is longer than 0.005 dS (Supplementary Fig. 4 of the manuscript).

3. The values for DNAJC11 look unreal. 39:1?? Really? Wow. To me this needs greater exploration. For example, what is the dN/dS ratio for this protein in other species: is the gene generally highly selected or does this represent a one-off event? It is also worth testing whether this represents positive selection for change or ‘normal’ selection plus a very high mutation rate. By this I mean it could be that any change to the amino acid sequence is beneficial (perhaps!). Alternatively, there could be huge numbers of changes due to a very high mutation rate, and of these a small subset is selected, but because of the high mutation rate, this is still a large number. In reality, I struggle to see how 39:1 can work because it implies that neutral, non-synonymous changes never hitchhike, which seems unlikely. In short, this ratio is so very extreme that I think it needs extra work to show that it is biologically plausible / even possible.

As shown in Table 1, the overall dN/dS ratio (excluding the edge with burst) for gammarids’ proteins, including DNAJC11 protein, is quite low (0.41 for DNAJC11), indicating that there is no excess of nonsynonymous substitutions in other parts of the phylogeny. This fact supports our suggestion that the bursts are limited to the particular short internal edges of the phylogenies (~0.001 dS).

This point is similar to point 2 made by Dr. Braun, and we added a short discussion of this monstrous burst to the text. We wish there were at least 10 of such events! Hopefully, they will be discovered when more data become available.

Reviewer: 2

We are very impressed by and most thankful for this meticulous review, which is almost as long as our manuscript. In particular, we are happy that Dr. Braun believes that we are not guilty of a number of sin, ascribed to us by several previous reviewers. We provide point-by-point response to numerous suggestions and ideas expressed there and also list the changes which we made after thinking of what Dr. Braun said.

Comments to the Author(s)

Overall, I am enthusiastic about the manuscript "Bursts of amino acid replacements in protein evolution" by Stolyarova and colleagues. I have long been interested in the topic of overdispersion in protein evolution and I believe the authors make a substantial advance in the field. I'd like to start my review by highlighting a few general concerns I have regarding the writing, then move on to specific concerns, and finish with a brief discussion of criticisms in the other reviews that do not concern me (i.e., areas where I feel the authors' work is solid and the criticisms are not valid).

The general criticism that I would articulate about the paper is the rather telegraphic style. Although the material that is there is well-written and clear, there is certainly room to expand the manuscript in a number of places. Royal Society Open Science does not have a word limit. Obviously, the absence of a word limit should not be viewed as a license to include irrelevant information. However, the current manuscript goes much too far in the opposite direction; I believe information that is necessary to understand and evaluate the science has been omitted.

With respect to the writing, I would like to start out with the introduction. The very first paragraph, which is critical to set up the contrast of gradual vs. punctuated evolution seems to be somewhat muddled. As it is currently written, the introduction jumps back and forth in terms of gradualistic and punctuated tempos for evolution and between molecular evolution and whole organism evolution. I think this is what reviewer 3 was responding to in their statement that "...the manuscript attempts to frame [the authors'] research in terms of punctuated evolution and stasis, which [reviewer 3] found much less relevant" than discussing the authors' work in the context of overdispersion.

Our goal in this paper is rather limited - we mostly wanted to describe the phenomenon of rapid bursts of evolution and to only sketch how it fits into the general context of research on the rates of evolution. Thus, we do not want to go too deep into the long history of debates between gradualists and punctualists. Nevertheless, a number of points made by Dr. Braun fit into our approach, so that we modified the manuscript accordingly.

Of course, the authors are correct when they responded to this by stating that there "...is no fundamental difference between overdispersed rate of evolution and alternating episodes of evolution and stasis." However, there can be many causes for alternating episodes of stasis and evolution and the causes of punctuation in molecular and morphological evolution. All of the examples of gradual evolution are molecular, despite the long history of using Brownian motion to model drift in morphological data (see O'Meara et al. 2006). If the authors are going to include examples of morphological evolution they acknowledge that gradual evolution of morphology is possible (indeed, it is often viewed as a null hypothesis).

Another aspect of the introduction that I found troubling is that all of the examples they cite to illustrate gradual evolution are molecular clock studies; to my knowledge molecular clock studies have not tested for overdispersion. Some overdispersed loci might be rejected indirectly when tests of the clock are performed, but modern relaxed clock studies might very well retain some loci that appear non-clocklike due to overdispersion. Studies that use molecular clocks (especially relaxed molecular clocks) are not evidence of gradual evolution per se, unless those studies test for overdispersion (i.e., punctuation) and reject it in favor of gradualism (at least for a subset of loci). Note that there are cases where the overdispersion hypothesis has been explicitly tested and rejected (e.g., Goldman 1994)

This brings us to the causes of punctuated evolution. At the morphological level it seems likely to me that most examples of punctuation reflect selection. This is really what the authors are looking for at the molecular level. However, there are causes of punctuation at the molecular level that reflect neutral evolution rather than selection. The authors acknowledge one of these explanations: gene

conversion. However, the authors do not appear to acknowledge that one of explanation for punctuated molecular evolution offered in the Pagel et al. (2006) study that they cite. Specifically, Pagel et al. (2006) stated that if "...speciation is associated with small founder populations and if genetic isolation is maintained, evolutionary rates can be accelerated at potentially all loci, because the number of loci with alleles governed by drift (neutral plus nearly neutral) is increased." In other words, Pagel et al. (2006) appear to favor a nearly neutral theory explanation for punctuation. Of course, Pagel et al. (2006) do offer selection as an alternative, stating that a "...second general mechanism for divergence is adaptive evolution as species invade new niches." The latter mechanism is the obviously what the authors hope to detect, but it is important that they set up the alternatives carefully.

We plagiarized one sentence from the review ("there can be many causes for alternating episodes of stasis and evolution and the causes of punctuation in molecular and morphological evolution.") because this idea cannot be expressed any better, and inserted it into the 1st paragraph of the Introduction.

We agree that gradualistic vs. punctualistic tempos of evolution must not be conflated with the level, molecular vs. morphological, at which evolution is observed. To avoid this, we now cite O'Meara et al. 2006. We also added a very appropriate citation of Gillespie 1991. In contrast, we do not want to open the Pandora box of the role of random drift in speciation (whatever this might mean) and would rather not comment in detail on the corresponding discussion in Pagel et al. 2006.

Overall, I really think the manuscript would be better served with an almost complete rewrite of the introduction. As I said, I agree completely that overdispersion is equivalent to alternating episodes of stasis and evolution. However, I did not feel that the introduction set up the analyses reported in the paper. Moreover, the extreme terseness of the introduction (and the manuscript as a whole) has led me to indicate to the editor that the language in its current form is not acceptable. I debated doing so since I usually state that language is unacceptable if there are statements I simply cannot understand. In this manuscript it is more the case that statements are too terse.

Anton Chekov famously said that "terseness is a sister of talent". We went over the whole text and identified several places where adding extra text is obviously warranted - and did so. However, we feel that our simple finding of 6 bursts of rapid evolution in two phylogenies is better served by a laconic paper.

In addition to these problems, I feel that the authors were somewhat sloppy with their citations. I have already brought up the issue that citing studies that use the molecular clock to estimate divergence times as evidence for gradualism is inappropriate; evidence for gradualism requires an actual test for overdispersion. The authors also cited the original Zuckerkandl and Pauling (1965) paper describing molecules as documents of evolutionary history in the context of selectively neutral segments of genomes; Zuckerkandl and Pauling (1965) predated neutral theory (Kimura 1968; King and Jukes 1969). I also fail to see how Cooney et al. (2017) represents an example of "gradual adaptive evolution"; after all, Cooney et al. (2017) state that their results "...support Darwinian⁹ and Simpsonian⁴ ideas of microevolution within adaptive zones and accelerated evolution between distinct adaptive peaks" (numbers in Cooney et al. 2017 quote are citations within that publication). The statement in Cooney et al. (2017) would seem to support a "burst" of substitution where taxa move from one adaptive peak to another. Finally, I was surprised by the failure to cite Gillespie (1991), which I believe to be the classic text describing overdispersed molecular evolution. Finally, the authors should make sure their references are formatted properly; I noticed that Moores et al. (1999)

is cited in the text as Ø Moores, Vamosi and Schluter, (1999); the authors should fix this and ensure there are no other mistakes regarding their citations.

We are thankful for this important comments on the citations. We do not understand what is wrong with our citation of Zukerkandl and Pauling (1965). Regarding Cooney at al. (2017), we thought that their data on the rates of bill evolution are consistent with gradualism. However, because Dr. Braun obviously knows more about birds than we do, it seems that we were wrong. Thus, were replaced this examples with two other examples of gradual adaptive evolution (Barrick et al., 2009; Mahler 2009). We also fixed the Moores et al. (1999) citations and checked other citations for mistakes.

Once I got past the introduction I felt the paper was much better and I generally became enthusiastic. The authors present evidence for adaptive overdispersed molecular evolution that I find quite compelling. I only have three issues with the current manuscript; I believe all three to be very important for the authors to address.

1. The issue of gene tree-species tree discordance.

Reviewer 3 also brought up the issue of discordance among gene trees, which the authors dismissed as not being a problem because they "...discarded the genes with poorly reconstructed ancestral sequences, which will be the case for discordant genes (e.g. caused by incomplete lineage sorting)." In other words, the authors reconstructed ancestral sequences using the species tree as the underlying topology and then examined posterior probabilities for the ancestral state reconstructions, discarding loci with low posterior probabilities. I am not at all convinced that poorly reconstructed ancestral sequences (as revealed by low posterior probabilities) are evidence of discordance due to incomplete lineage sorting (ILS).

I will illustrate my assertion that "codeml posterior probabilities for the reconstructed ancestral variant < 0.8 " will not necessarily highlight loci with discordant gene trees using two scenarios. First, let us imagine a gene encoded by a locus that has a gene tree that is perfectly congruent with the species tree and further postulate that the gene is rapidly evolving. I would expect the high degree of homoplasy that I postulated in this scenario to lead to low codeml posterior probabilities. Second, let us postulate a relatively slowly evolving gene associated with a locus that has a gene tree with some discordance due to ILS. The ILS could lead to some sites in the alignment that appear homoplastic (the appearance of homoplasy given the species tree due to gene tree-species tree discordance is called hemiplasy; see Avise & Robinson 2008) and the ancestral state reconstructions of those sites those might have reduced posterior probabilities. In this example the ancestral state reconstructions may also be incorrect given that they reflect hemiplasy that is misinterpreted as homoplasy, but I do not think it is a given that the reconstructions will definitely have a posterior probability < 0.8 .

Note that this discussion is not a criticism of the chosen posterior probability cut-off; it is an intrinsic problem with the interpretation of ancestral state reconstructions that have low posterior probabilities as reflecting discordance due to ILS. Basically, there are multiple ways for codeml ancestral state reconstruction to have low posterior probabilities for the reconstructed ancestral variants. Discordance due to ILS is likely to be one way since hemiplasy does create the appearance of homoplasy given the species tree. However, unless the authors have done simulations showing that low posterior probabilities in codeml reconstructions are definitely indicative of discordance I do not believe they should state they are.

Does this have a practical impact on the analyses that the authors present? I am uncertain whether it does because I am uncertain how much discordance among gene trees exists in the datasets the

authors examined. A simple way for the authors to assess discordance is to look for incongruent gene trees. I am well aware that error in gene tree estimation is a problem for estimating the amount of incongruence among gene trees (for examples of my publications in the area see Patel et al. 2013 and Meiklejohn et al. 2016). However, there is a simple solution for the authors: estimate gene trees with support values using a standard method, such as maximum likelihood (ML) bootstrapping using RAxML (Stamatakis 2014) or IQ-TREE (Nguyen et al. 2015), collapse all branches with less than some amount of support (e.g., 75% standard bootstrap support), and only consider branches with support when examining incongruence. There are important statistical issues with this approach, since there will be some errors (i.e., some branches will appear to be incongruent when they are not). However, it is a better criterion than using posterior probabilities of ancestral state reconstructions because the number of branches with support that conflict with the species tree is a direct measure of incongruence. Ideally, the authors would present this information for all genes (i.e., XX% of the branches in each gene tree received >75% bootstrap support; YY% of those branches conflicted with the species tree) but they should present this information for the cases where they found significant evidence for bursts.

I actually doubt that this will change their conclusions. It seems likely to me that the bursts define branches that are present in the gene tree. For example, if the 39 non-synonymous substitutions in the amphipod DNAJC11 sequences do not define a clade in the DNAJC11 gene tree then one has to postulate a truly astounding degree of convergence that also happens to appear to be synapomorphic changes when mapped on the species tree. Of course, the DNAJC11 gene tree could be discordant with the species tree elsewhere, but that doesn't matter from the standpoint of inferring bursts of substitutions.

Overall, I am not convinced that gene tree discordance is a problem but the way that the authors address it is not appropriate. At a minimum they should infer ML phylogenies for each gene with a burst and ask whether the branch in the species tree with the burst is present in the gene tree. There are four possible outcomes for this analysis of the gene tree: 1) the branch with the burst is present in the gene tree and supported by the ML bootstrap; 2) the branch with the burst is present in the gene tree but it is not supported by the ML bootstrap; 3) a branch that conflicts with relevant branch in the species tree is present but that conflicting branch is not supported by the ML bootstrap; and 4) a branch that conflicts with relevant branch in the species tree is present and the conflicting branch is supported by the ML bootstrap. It is only in case #4 that discordance can be said to be a potential problem. The more general analyses of incongruence that I mention above would be interesting, but it is not critical for the authors to prove their point.

We are thankful for these detailed suggestions and comments on our findings. Because we were studying rather rare events, we were concerned with the possibility of false positive findings. Thus, we used rather stringent filtering, as far as discordance between gene and species trees is concerned. Those bursts that we report occur in gene with absolutely no evidence of discordance. It is likely that there is some number of bursts that we filtered out which are real. However, we would rather have a smaller number of unambiguous cases. When the data on genomes, instead of transcriptomes, of the lake Baikal amphipods become available (unfortunately, their genomes are rather large, so this is probably going to wait for several more years), we hope to redo the analysis using the better data.

2. The puzzle underlying the mutational basis for their extreme cases of overdispersion

I believe that DNAJC11 illustrates something that may be even more interesting than the scenario that the authors postulate. I'd like to start by considering two limits on the rate of accumulation of

non-synonymous substitutions given that the authors state that the relevant branch in the species tree has a very short synonymous length ($dS = 0.004$).

Let's begin with what I'll call the "neutral limit." There are there are 1515 sites in the alignment (I am using gap free sites, but the total number of sites is not much larger). If we assume the number of synonymous sites is $\sim 1/3$ of the total number of the expected number of synonymous substitutions for that branch is ~ 2 . If we assume synonymous sites are selectively neutral the rate of accumulation for synonymous substitutions is the mutation rate (Kimura 1968). Obviously, many non-synonymous sites will be subject to purifying selection but this dS would imply that, if all non-synonymous sites were selectively neutral (i.e., if $dN/dS = 1.0$), that the expected number of non-synonymous substitutions is only ~ 4 . However, the authors observed 39 non-synonymous substitutions in this locus; the probability of this many non-synonymous substitutions when the expected number of substitutions if we assume a Poisson process is extremely low. Of course, the authors would reply "that is our point" to this description of the "neutral limit." However, my point in articulating this to place a lower limit on the number of non-synonymous substitutions that it is reasonable to expect if selection for changes at individual sites is relatively weak; the expectation given neutrality provides that lower limit. I would add that this "neutral limit" is really just a restatement of the authors' null hypothesis; in fact, they stated that they "...assumed the neutral null model ($dN=dS$)" and emphasized that this will make "...it possible to detect only the longest and the fastest[sic] bursts."

Another way of articulating the fact that the authors' method can "...detect only the longest and the fastest[sic] bursts" is to say that the non-synonymous substitutions must be subject to strong selection. However, this need for strong selection on new non-synonymous variants brings us to a different limit that I'll call the "Hill-Robertson limit." If we assume the selection for individual non-synonymous changes is strong then those alleles should be fixed rapidly. This would tend to eliminate other alleles, even if those other alleles are advantageous relative to the ancestral allele. This is just a restatement of the well-known Hill & Robertson (1966) effect. Given Hill-Robertson interference one must assume that a large number of potential non-synonymous changes become advantageous and then occur successively. In other words, Hill-Robertson interference requires one to postulate the following scenario: 1) a new advantageous mutation enters the population; 2) the new mutation becomes fixed in the (metaphorical) blink of an eye; 3) another advantageous mutation enters the population (in the background of newly fixed advantageous allele); and 4) this cycle repeats for a total of 39 times. Moreover, even if one imagines an environmental shift occurs that favors specific substitutions at >30 sites one would think that epistatic interactions among sites within the protein would create issues (though I suppose one could also argue that epistasis might create new "opportunities" for changes that are subject to positive selection). Perhaps there are parts of parameter space (i.e., mutation rate, generation time, effective population size, selection coefficients on individual sites, etc.) where this can occur. But it seems to me that the "good" part of parameter space might be very small.

I can see two ways out of the narrow space created by these two limits: 1) recombination rates are high enough to overcome Hill-Robertson interference; or 2) the mutation that generated these non-synonymous changes altered multiple sites. The first explanation would probably require that there is a reasonable rate of recombination between sites that are ~ 10 bp apart (some of the amino acid substitutions uniting the group defined by shading in Fig. 1 are 2-3 amino acids apart). The second might imply that only a subset of the non-synonymous substitutions are advantageous; the remainder were simply "dragged" to fixation by the advantageous change(s). Both possibilities are interesting. It is debatable whether this argument applies to all of their examples, but the authors were looking for exceptional outcomes. Perhaps the exceptional outcomes indicate that surprising molecular mechanisms exist.

Finally, the authors should fix the misspelling of “fastest” indicated above using “[sic]”.

We agree - this monstrous burst rises interesting issues. Assuming the per nucleotide per generation mutation rate $\sim 10^{-8}$ (observed in a number of animals), an edge with $dS = 0.004$ corresponds to 400,000 generations. This leaves $\sim 10,000$ generations for each of our 39 substitutions - not much, because, in the course of this time, a mutation must occur, must be lucky enough to avoid early random extinction (with probability $2s$), and must be carried to fixation by positive selection. Perhaps, this suggests that, from the very beginning of a burst, selection favored new alleles at many sites, and the actual substitutions occurred in the order dictated by availability of lucky mutations. Indeed, the probability of recombination bringing together two very tightly linked advantageous substitutions at such a short timescale is low.

We add this reasoning to the discussion. However, a single monstrous event is probably not enough grounds for developing a quantitative theory. Hopefully, more such bursts will be discovered soon - and, as Dr. Braun suggests, they may shed light on how positive selection actually works.

It is, however, very unlikely that bursts are due to complex mutations, because a complex mutation would affect both nonsynonymous and synonymous sites.

3. The base composition of synonymous sites

dS can be underestimated when there is strong codon bias (Rabinowicz et al. 1999). The authors should present the base composition (and variation among loci in their base composition) for the data used to calculate dS .

We do not think that there could be a problem here. True, a strong codon bias may affect estimates of dS . However, it can happen only when we are dealing with substantially different sequences, such that multiple nucleotide replacements at a site were routine in the course of their divergence from the most recent common ancestor, and estimating their number is the key step in determining dS . Fortunately, our task is much simpler: we need to ascribe dS values to extremely short edges of phylogenetic trees, with extremely low probability of multiple events at a site. Thus, as long the sequences that correspond to the internal nodes of the phylogenetic tree are inferred correctly (and we ignored cases where PAML did not infer them with confidence, see Methods), lengths of edges must also be correct.

4. The location of non-synonymous substitutions

The secondary structure of the Catarrhini PKR sequences is shown but the secondary structure of the DNAJC11 sequences is not shown. I'm not convinced that presenting the secondary structure is important unless other information is provided, but the authors should be consistent. It is more important to indicate the locations of the substitutions relative to the boundaries of protein domains, which the authors do for PKR. It would be good to do this for DNAJC11 as well; for example, how many of the 39 substitutions are located in the J domain?

Finally, there are two words that should be fixed in the Fig. 3 legend: “others” and “in”. I have indicated them in CAPS below:

“Fig. 3. Alignment of PKR genes of Catarrhini (fragments) containing 18 nonsynonymous substitutions on the internal edge ancestral to *Macaca mulatta* and *Macaca fascicularis*. The majority of

substitutions (14) occurred in the kinase domain of the protein, the OTHERS fall IN the dsRNA binding motif (DRBM1). Alleles derived in the adaptive burst are shown in bold.”

Unfortunately, the most protein sequences of Baikal Lake amphipods are hard to annotate, that’s why we didn’t show detailed structure of these proteins. For example, the most part of the DNAJC11-called gene doesn’t show homology to any domains of known function. The DnaJ domain, which is associated with hsp70 heat-shock chaperone system, covers only ~80 aa sites on the N-terminal end of the sequence. This domain has only one substitution that can be possibly referred to the burst, however, we didn’t include it in our analysis due to gaps. So, we can’t do any conclusions about how the burst-composing substitutions are distributed relative to protein domains or secondary structure. We now fixed the error in Fig. 3 legend.

5. Data accessibility

I was unable to access the Dryad data package. Assuming that it is made available and includes (at a minimum) the following:

- 1) Nucleotide sequence alignments of the loci they examined (ideally all sequences) in a readily usable format (e.g., nexus or relaxed phylip).
- 2) The phylogenetic trees they used for their analyses in newick format.
- 3) The gene names and functional annotations (especially the blast2GO data for the amphipods).

I think it will be appropriate.

We regret that Dr. Braun couldn’t access the data deposited in Dryad, it should be accessible by direct link <https://datadryad.org/resource/doi:10.5061/dryad.40vp30d> . It contains both nucleotide sequence alignments of burst-containing genes (in fasta format) and phylogenetic trees. The blast2GO annotations for the genes of the amphipods were published in the original paper (Naumenko 2017, Table S8). Not our fault we hope - we just followed the instructions.

I would like to emphasize, despite the fact that I wrote a long review pointing out issues, that I am enthusiastic about this manuscript. I think the authors have presented good evidence for their central assertion that there were adaptive bursts occur in small numbers of proteins. Thus, I believe the science is sound. I also think their conclusions are of broad interest.

I hope the authors and editor find these comments helpful.

Edward L. Braun
Professor of Biology
University of Florida
Gainesville, FL 32611

REFERENCES:

Awise JC, Robinson TJ. 2008. Hemipty: a new term in the lexicon of phylogenetics. *Systematic Biology*, 57(3): 503-507. DOI: 10.1080/10635150802164587

Gillespie JH. 1991. *The Causes of Molecular Evolution* (Oxford Series in Ecology and Evolution, edited by May RM and Harvey PH). Oxford University Press, Oxford. ISBN-10: 0195092716

- Goldman N. 1994. Variance to mean ratio, $R(t)$, for Poisson processes on phylogenetic trees. *Molecular Phylogenetics and Evolution*, 3(3): 230-239 DOI: 10.1006/mpev.1994.1025
- Hill WG, Robertson A. 1966. The effect of linkage on limits to artificial selection. *Genetical Research*, 8: 269–294 DOI: 10.1017/S0016672300010156
- Kimura M. 1968. Evolutionary rate at the molecular level. *Nature* 217: 624-626 DOI:10.1038/217624a0
- King JL, Jukes TH. 1969. Non-darwinian evolution. *Science*, 164(3881): 788-798 DOI: 10.1126/science.164.3881.788
- Meiklejohn KA, Faircloth BC, Glenn TC, Kimball RT, Braun EL. 2016. Analysis of a rapid evolutionary radiation using ultraconserved elements: evidence for a bias in some multispecies coalescent methods. *Systematic Biology*, 65(4): 612-627. DOI: 10.1093/sysbio/syw014
- Nguyen L-T, Schmidt HA, von Haeseler A, Minh BQ. 2015. IQ-TREE: A fast and effective stochastic algorithm for estimating maximum likelihood phylogenies. *Molecular Biology and Evolution*, 32: 268-274 DOI: 10.1093/molbev/msu300
- O’Meara BC, Ané C, Sanderson MJ, Wainwright PC. 2006. Testing for different rates of continuous trait evolution using likelihood. *Evolution*, 60(5): 922-933. DOI: 10.1111/j.0014-3820.2006.tb01171.x
- Pagel M, Venditti C, Meade A. 2006. Large punctuational contribution of speciation to evolutionary divergence at the molecular level. *Science*, 314(5796): 119-21. DOI: 10.1126/science.1129647
Erratum in: *Science*, 2006 314: 925.
- Patel S, Kimball RT, Braun EL. 2013. Error in phylogenetic estimation for bushes in the tree of life. *Journal of Phylogenetics and Evolutionary Biology* 1: 110 DOI: 10.4172/2329-9002.1000110
- Rabinowicz PD, Braun EL, Wolfe AD, Bowen B, Grotewold E. 1999. Maize R2R3 Myb genes: sequence analysis reveals amplification in the higher plants. *Genetics*, 153(1): 427-444
- Stamatakis A. 2014. RAxML version 8: a tool for phylogenetic analysis and post-analysis of large phylogenies. *Bioinformatics*, 30(9): 1312-1313 DOI: 10.1093/bioinformatics/btu033

Appendix B

Dear Dr Steve Brown,

Thank you for accepting our manuscript "Bursts of amino acid replacements in protein evolution" (ID RSOS-181095) to publication in Royal Society Open Science with minor revision. We carefully reviewed the comments from the referee and responded to them below.

**Best regards,
Anastasia Stolyarova
(on behalf of all authors)**

07-Feb-2019

Dear Ms Stolyarova:

On behalf of the Editors, I am pleased to inform you that your Manuscript RSOS-181095.R1 entitled "Bursts of amino acid replacements in protein evolution" has been accepted for publication in Royal Society Open Science subject to minor revision in accordance with the referee suggestions. Please find the referee's comments at the end of this email.

<...>

on behalf of Dr Steve Brown (Subject Editor)
openscience@royalsociety.org

Associate Editor Comments to Author (Dr Steve Brown):

Thank you for the many detailed responses, which will now be considered by the reviewers of your paper.

Reviewer comments to Author:

Reviewer: 1

Comments to the Author(s)

I feel my concerns have largely been addressed. However, I still worry about the 'monstrous hit'. This is such an extreme case that it needs to be treated with great caution, particularly since it is mitochondrial-associated, which raises the possibility of contamination from homologous transcripts from related genes (nuclear vs mtDNA copies / paternally inherited mtDNA etc.). I realise that there has been filtering by dS value but wonder whether this is enough. Mitochondrial sequences have highly polarised based frequencies which could greatly reduce dS values relative to a naive JK model. For me, more analysis is needed, for example by including a tree based synonymous substitutions for the gene that will hopefully be concordant with the overall tree, yet diverge strongly from an equivalent tree based on non-synonymous changes. Extreme observations require extreme support before they can be taken seriously. This is particularly true when the extreme observation is one of very few similar observations.

We thank the reviewer for the detailed comments on our most extreme finding. We checked that base composition of this gene and its is not biased (26% A, 21% T, 26% C, 27% G), suggesting that the gene is not of recent mitochondrial origin. According to the advice of the reviewer, we reconstructed the phylogeny of this gene based on fourfold degenerate sites only (shown on the figure below, numbers indicate bootstrap values for the corresponding branches). Except for the branches with low bootstrap values, this tree is concordant with the overall *Pallasea* gammarids phylogeny (Fig. S1 of the manuscript); the burst-containing branch is also reproduced with high bootstrap value. Hence there is no evidence of the contamination of any kind. Still, we are reluctant to add this figure to the paper due to many low bootstraps - which, in turn, are due to a small number of substitutions at fourfold degenerate sites within this gene.

Journal Name: Royal Society Open Science

Journal Code: RSOS

Online ISSN: 2054-5703

Journal Admin Email: openscience@royalsociety.org

Journal Editor: Emilie Aime

Journal Editor Email: emilie.aime@royalsociety.org

MS Reference Number: RSOS-181095.R1

Article Status: SUBMITTED

MS Dryad ID: RSOS-181095.R1

MS Title: Bursts of amino acid replacements in protein evolution

MS Authors: Stolyarova, Anastasia; Bazykin, Georgii; Neretina, Tatyana; Kondrashov, A

Contact Author: Anastasia Stolyarova

Contact Author Email: anastasia.v.stolyarova@gmail.com

Contact Author Address 1:

Contact Author Address 2:

Contact Author Address 3:

Contact Author City: Skolkovo

Contact Author State:

Contact Author Country: Russian Federation

Contact Author ZIP/Postal Code: 143025

Keywords: punctuated equilibrium, rate of evolution, adaptive walks, positive selection

Abstract: Evolution can occur both gradually and through alternating episodes of stasis and rapid changes. However, the prevalence and magnitude of fluctuations of the rate of evolution remains obscure. Detecting a rapid burst of changes requires a detailed record of past evolution, so that events that occurred within a short time interval can be identified. Here, we use the phylogenies of the Baikal Lake amphipods and of Catarrhini, which contain very short internal edges facilitating this task. We detect 6 radical bursts of evolution of individual proteins during such short time periods, each involving between 6 and 38 amino acid substitutions. These bursts were extremely unlikely to have occurred neutrally, and were apparently caused by positive selection. On average, in the course of a time interval required for one synonymous substitution per site, a protein undergoes a strong burst of rapid evolution with probability at least ~ 0.01 .

EndDryadContent